# Monitoring of Heart Rate from Photoplethysmographic Signals Using a Samsung Galaxy Note8 in Underwater Environments

**DOI:** 10.3390/s19132846

**Published:** 2019-06-26

**Authors:** Behnam Askarian, Kwanghee Jung, Jo Woon Chong

**Affiliations:** 1Department of Electrical and Computer Engineering, Texas Tech University, Lubbock, TX 79409, USA; 2Educational Psychology and Leadership, Texas Tech University, Lubbock, TX 79409, USA

**Keywords:** photoplethysmography, PPG, heart rate variability, underwater, temperature

## Abstract

Photoplethysmography (PPG) is a commonly used in determining heart rate and oxygen saturation (SpO_2_). However, PPG measurements and its accuracy are heavily affected by the measurement procedure and environmental factors such as light, temperature, and medium. In this paper, we analyzed the effects of different mediums (water vs. air) and temperature on the PPG signal quality and heart rate estimation. To evaluate the accuracy, we compared our measurement output with a gold-standard PPG device (NeXus-10 MKII). The experimental results show that the average PPG signal amplitude values of the underwater environment decreased considerably (22% decrease) compared to PPG signals of dry environments, and the heart rate measurement deviated 7% (5 beats per minute on average. The experimental results also show that the signal to noise ratio (SNR) and signal amplitude decrease as temperature decreases. Paired *t*-test which compares amplitude and heart rate values between the underwater and dry environments was performed and the test results show statistically significant differences for both amplitude and heart rate values (*p* < 0.05). Moreover, experimental results indicate that decreasing the temperature from 45 °C to 5 °C or changing the medium from air to water decreases PPG signal quality, (e.g., PPG signal amplitude decreases from 0.560 to 0.112). The heart rate is estimated within 5.06 bpm deviation at 18 °C in underwater environment, while estimation accuracy decreases as temperature goes down.

## 1. Introduction

Cardiovascular diseases (CVDs) are considered one of the main reasons of deaths causing more than 30% of deaths around the world [1]. It is reported that 2200 out of 801,000 deaths in the U.S are caused by CVDs. Almost 47% of sudden CVD deaths occur outside the hospital, which indicates that most individuals may not manage warning signs of heart disease [2,3]. Accordingly, continuous heart rate monitoring is highly demanded to prevent the complications of CVDs. Recently, smartphone applications for healthcare, e.g., mobile heart disease detection, heart rhythm analysis, remote home care monitoring, eye disease diagnosis, have become highlighted [4,5,6,7,8,9,10]. Moreover, smartwatches, wrist bands, and activity trackers based on electrocardiography (ECG) or photoplethysmography (PPG) are also widely used for continuous measurement of physiological signs such as heart rate and peripheral oxygen saturation (SpO_2_) in daily life [11]. Among these technologies, PPG using a smartphone camera or smartwatches is highlighted as continuous and comfortable method for heart rate measurement since it does not require additional electrodes or skin preparation. Moreover, PPG measurement using smart devices was shown to be effective in detecting arrhythmia [12,13,14].

PPG signals can reflects changes in blood volume below underlying body tissues in a simple, noninvasive, and low-cost manner [15]. PPG sensor technology facilitates the development of several wearable devices such as smartwatches, Fitbits, and activity trackers for easy monitoring of heart rate in rest state, during daily activities, or exercises [13]. Since PPG measurement is sensitive to any disturbance and is affected by changes in the reflected light intensity induced by the variations in the volume of blood [13], motion and noise artifacts (MNA) caused by any movement or physical activity, displacement of the finger, finger pressure, and temperature can cause disturbance and discrepancy in PPG measurements, which yields to inaccurate heart rate measurements [16]. Therefore, there have been studies improving PPG signals acquired in the presence of MNA [17,18]. The effects of skin and room temperature on PPG signals have been studied in [19], which shows that cold temperature significantly reduces the amplitude of PPG signals and the accuracy of the SpO_2_ measurement while those are improved in warm temperature.

Representative examples of constant heart rate monitoring during underwater activities are (1) heart rate monitoring of divers [20], and (2) heart rate monitoring of subjects during water walking or jogging, which is suggested for rehabilitation and fitness enhancement. Underwater electrodes have been proposed using carbon black powder (CB) and polydimethylsiloxane (PDMS) for underwater ECG measurements [21]. Cold water is observed to influence both heart rate and blood pressure in [22]. Changes in blood pressure of trained divers during cold water immersion was analyzed by asking the divers to hold their breath during the cold water submersion in [20]. The diver’s vital signs were analyzed using an ECG unit and the study focused on the biological changes in the body that effects the changes in blood pressure. The influence of cold water immersion on blood flow has been investigated [22] while The effects of underwater activities (water immersion, submersion, and scuba diving) on heart rate variability has been studied [23]. Comparison of surface electromyography (sEMG) electrodes for measuring muscle movement in water and land has been studied [24] and Electromyography (EMG) sensors measuring muscle movement in rehabilitation treatment in exercise pools were studied in [25,26]. However, these studies do not use PPG but ECG which requires additional electrodes.

In this paper, we evaluate the performance of PPG sensors embedded in the Samsung Galaxy Note8 in dry and underwater environments. As performance metrics, we consider the accuracy of the heart measurements, signal amplitude, and signal to noise ratio (SNR). Specifically, we compared the acquired information from PPG signals to a gold-standard reference which is obtained from NeXus-10 MKII (Mind Media, Herten, Germany) device. The NeXus-10 MKII has a Food and Drug Administration (FDA) approval. The effect of cold temperature on PPG signals was studied in the viewpoints of (1) physiology and (2) sensor hardware. In the viewpoint of physiology, the effect of cold temperature on PPG signals was analyzed in terms of blood vessels in human body [27,28,29]. In this study, a temperature drop from 20 °C to 3 °C was observed to increase heart rate from 82 to 98 beats per minute (bpm) [28,29], and increase oxygen saturation level from 97% to 99%. Moreover, the temperature drop caused blood viscosity to increase, which decreased the PPG signal amplitude. On the other hand, the effect of cold temperature on PPG signal amplitude was studied in the viewpoint of sensor hardware including its photodiodes [19]. For example, temperature change from 25 °C to 4 °C shifted the wavelength by 18 nm, decreased the voltage by 0.1 volts, and decreased the current by 0.05 amps. However, cold temperature was observed not to change the pattern (or shape) of PPG signals in both physiology and sensor hardware viewpoints [30]. The rest of this paper is organized as follows. Section 2 describes data collection, measurement devices, and study protocol. In Section 3, our proposed image processing method consisting of signal extraction, data processing, feature extraction, and heart rate measurement is explained. Experimental results are presented in Section 4. Finally, Section 5 concludes this paper.

## 2. Materials

### 2.1. Measurement Devices

In this paper, we used FLUKE SPOT Light Functional tester which generates artificial PPG signals for calibration purpose, Samsung Galaxy Note8 for data acquisition and Nexus-10 MKII for gold-standard reference which are shown in Figure 1a–c respectively. Samsung Galaxy Note8 is a water- and dust-resistant smartphone which has a PPG sensor on the rear side for the purpose of measuring heart rate and SpO_2_ [31]. The NeXuS-10 MKII is equipped with sensors measuring physiological signals including electroencephalogram (EEG), ECG, PPG, SpO_2_, respiration signals, and EMG [32]. When measuring PPG signals, Samsung Galaxy Note8 uses a reflection mode while the NeXus-10 MKII device uses a transmission mode. Both devices use a 645 nm red light emitting diode (LED) and a 940 nm infra-red LED.

### 2.2. Data Acquisition

We recruited 20 healthy volunteers following the Texas Tech University (TTU) Institutional Review Board (IRB) (IRB#: IRB2018-688). The ages of volunteers were ranged from 18 to 80 years old. The volunteers were asked to contact the index finger of the left hand on the NeXus-10 MKII device and the index finger of the right hand on the Samsung Galaxy Note8 smartphone. This hand selection was not randomized but the same for all volunteers. The volunteers were asked to sit in a relaxed position, breathe normally, and avoid any movement for the duration of data acquisition. The position of the volunteer’s hand was in a stable and relaxed position on a table at the height of their heart. In this paper, we considered (1) underwater and (2) dry environments. The underwater and dry experiments were performed serially without any interruption since heart rate can change due to stress, activity, and even due to changes in the volunteer’s position. The calibration process was done on each volunteer before each data acquisition, and we used SPOT light pulse oximeter analyzer [33] shown in Figure 1a to generate artificial PPG signals. The calibration was performed to validate that smartphones and NeXus devices are working correctly before the actual measurement. The length of each measurement varied due to the time it takes for the smartphone application to get cleaner PPG signals, which is required for the Samsung Galaxy Note8 app’s heart rate estimation.

a. Dry environment

In the dry environment, the volunteers were instructed to hold the smartphone with covering the Samsung’s PPG sensor by the index fingertip of their right hand while the NeXus-10 MKII was attached to the index fingertip of their left hand. The beginning and end time of the measurement and the calculated heart rate by the smartphone built-in application was recorded for our analysis. The data acquisition setup for dry environment is shown in Figure 2a.

b. Underwater environment

In this environment, we performed two different experiments: (1) Underwater Samsung smartphone vs dry NeXus device at 18 °C fixed temperature, and (2) Samsung smartphones in underwater various temperatures. In the first experiment, to eliminate temperature factor and focus only on the medium factor, we compare dry and underwater (Samsung) PPG signal at a fixed temperature as shown in Figure 2. For underwater experiment, volunteers were asked to hold the smartphone in an emulated underwater condition while the alternate index finger with the NeXus PPG sensor was out of the water for the baseline measurement as shown in Figure 2b. The underwater condition is emulated by a transparent bucket of water containing clear water with the temperature maintained at the room temperature (18 °C) as shown in Figure 2b. The temperature of the water was set as the room temperature (18 °C) to eliminate the effect of temperature variations on the PPG measurement.

In the second experiment, we varied the temperature in three steps: warm, moderate, and cold. Specifically, we considered 45 °C, 18 °C and 5 °C for warm, moderate, and cold temperature, respectively. The water was first heated up to 45 °C and the volunteer was asked to immerse their finger in the warm water. We add ice to cool the water down. It took around 20 s for the temperature of water goes down by 1 °C. We first measured data at 45 °C for the warm temperature and put ice into the water. After water temperature reaches 18 °C, the data at were measured for the moderate temperature. After water temperature reaches 5 °C, the data were measured at cold temperature. These tests were performed in a serial way to minimize the changes of a volunteer’s heart rate.

## 3. Method

### 3.1. Data Processing

We extracted raw PPG signals from the PPG signal images on the smartphone screen by our proposed image and signal processing algorithms. Figure 3 shows the flowchart of our proposed data processing procedure which consists of acquiring screenshot during the measurement, cropping acquired images, converting RGB (Red, Green, Blue) image to HSV (Hue, Saturation, Value) color space, obtaining a binary image after denoising background from HSV images, and scaling a binary image to values in x-y axis. We implement this data processing procedure using MATLAB 2017b (from MathWorks) [34].

First, the PPG signal images on the smartphone screen were acquired by the built-in application for the PPG sensor embedded in Galaxy Note8 smartphone. During the time of measurement, the PPG signal was shown on the screen of the smartphone and the final output of the application is estimated heart rate value. To evaluate the quality of the raw PPG signals obtained by Galaxy Note8, we needed to extract the raw PPG signal from the PPG signal images on the smartphone screen. We used the Mobizen screen recording application which provides full high-definition (HD) smartphone screen recording at 60 frames per second (fps) [35]. PPG signal extraction from the smartphone started with image cropping from recorded videos of smartphone screen by this Mobizen application. We then applied the stitching technique to align consecutive screenshots after the screenshots were acquired [36,37]. 

After obtaining a stitched RGB PPG image, we convert this RBG image into HSV color space image. Then, noise and background were reduced in the HSV images by the color mask technique of our proposed method [38]. Finally, the denoised image was converted to a binary image which itself is the raw PPG signal that we aimed to obtain. For the heart rate calculation from the raw PPG signal, we use the maximum amplitude value at each pulse (or heartbeat) of the raw PPG signal. The detailed procedure converting from RGB images to heart rate (RGB 🡪 HSV 🡪 color masking 🡪 binary 🡪 scaling 🡪 heartbeat calculation) is described in Section 3.2.

### 3.2. PPG Signal Extraction from RGB Images

Figure 4a shows an example of a smartphone screenshot whose size is 1080 × 2220 pixels. First, we cropped the image to extract only the region in which the signal exists (see Figure 4b). Next, the image of the signal in the RGB color space was converted to the HSV color space using the HSV color conversion method. The reason why we used the HSV color space was that HSV color space is shown to be effective color space in detecting signals from bright background [38]. By defining the thresholds for each of the channels of the image based on the histograms of the S and V components, the binary image of the signal was extracted from its background. The reason why we chose S and V component was that S and V components give larger difference values between signal and background compared to H component. For S and V components, we set the thresholds to be 0.6 and 0.7, respectively. If V values are lower than these thresholds, then it is mapped into *black*. Otherwise, it is mapped into *white*. The points in the binary image of the signal, which are shown in Figure 4c, were then scaled to the *x-y* axis to construct PPG signals in time domain as shown in Figure 4d. The constructed PPG signals from the screenshots were stitched together, and finally, the heart rate is extracted from the stitched PPG signal (see Section 3.3).

Specifically, the HSV color conversion method is described in Equations (1) to (5). First, the RGB values are normalized using the following equation:(1)$$R^{\prime }=R/255,\;\;G^{\prime }=G/255,\;\;B^{\prime }=B/255.$$
where $R^{\prime }$, $G^{\prime }$, and $B^{\prime }$ are the normalized values of the RGB color space and each of the values is in the range of 0 to 1.

The maximum ($C_{max}$) and minimum ($C_{min}$) values of the normalized RGB channels ($R^{\prime }G^{\prime }B^{\prime }$) and the difference Δ between the maximum and the minimum values of the normalized RGB channels are calculated as follows:(2)$$C_{max}\;=\;\max\left(R^{\prime },G^{\prime },B^{\prime }\right),\;C_{min}\;=\;\min\left(R^{\prime },G^{\prime },B^{\prime }\right),\;\Delta \;=\;C_{max}-C_{min}.$$
The Hue (H) component of the HSV color space image is calculated from the following equation:(3)$$H=\{\begin{array}{c} 0\;\;\;\;\;\;\;\;\;\;\;\;\;\;\;\;\;\;\;\;,\;\mathrm{if}\;\Delta =0\\ 60{}^{\circ}\times (\frac{G^{\prime }-B^{\prime }}{\Delta })\;\;\;\;\;\;,\;\mathrm{if}\;C_{m}{}_{ax}=R^{\prime },\\ 60{}^{\circ}\times (\frac{B^{\prime }-R^{\prime }}{\Delta }+2),\;\mathrm{if}\;C_{m}{}_{ax}=G^{\prime },\\ 60{}^{\circ}\times (\frac{R^{\prime }-G^{\prime }}{\Delta }+4),\;\mathrm{if}\;C_{m}{}_{ax}=B^{\prime }. \end{array}$$
The saturation (S) component of the HSV color space image is calculated by implementing Equation (4):(4)$$S=\{\begin{array}{c} 0\;\;\;\;\;\;,\;C_{max}=\;0,\\ \frac{\Delta }{C_{max}}\;\;,\;C_{max}\neq 0. \end{array}$$
Finally, the value (V) component of the HSV color space image is derived from the following equation:(5)$$V=C_{max}.$$

### 3.3. Signal Quality Index (SQI) and Heart Rate Calculation and from the PPG Signal

To evaluate the effect of underwater conditions on heart rate measurement and compare the results with those from the dry environment, we derived SQI as well as heart rate from PPG signals. As SQI, we consider signal amplitude and SNR [39]. Signal amplitude $A_{\mathrm{signal},\;n}$ at the *n*^th^ pulse is calculated by $A_{\mathrm{signal},n}=S_{\mathrm{peak},\;\;n}\;-\;S_{\mathrm{trough},n}$ where $S_{\mathrm{peak},\;\;n}\;$ and $S_{\mathrm{trough},n}$ denote sampled values of peak and trough at the *n*^th^ pulse. SNR compares the signal level to its background noise and is calculated by:(6)$${\mathrm{SNR}}_{\mathrm{db}}\;=\;20\log\left(\frac{A_{\mathrm{signal}}}{A_{\mathrm{noise}}}\right),$$
where $A_{\mathrm{signal}}$ is signal amplitude and $A_{\mathrm{noise}}$ is noise amplitude.

In cold temperatures, peripheral blood flow reduces due to vasoconstriction of the blood vessels, which leads to a smaller PPG signal amplitude [28]. According to Snell’s Law, on the other hand, any medium change alters the light speed causing a deviation in the transmitted and received light beam [40]. Snell’s Law declares that the speed of the light traveling from one medium (air) to another medium (water) will change based on the following equation which results in changes in the light direction:(7)$$\frac{\sin\theta_{1}}{\sin\theta_{2}}\;=\;\frac{\vartheta_{1}}{\vartheta_{2}}\;=\;\frac{n_{1}}{n_{2}},$$
where $\theta_{1}$ is the angle of incidence in air environment, $\theta_{2}$ is the angle of refraction in the water environment, $\vartheta_{1}$ is the speed of light in dry environment, and $\vartheta_{2}$ is the speed of light in the underwater environment as shown in Figure 5. As a result, deviation in the light beam of the PPG sensor makes less light enter the PPG photodetector sensor, which results in lower signal amplitude. Moreover, the water absorbs some of the light, which may also decrease the amount of light entering the photodetector.

Heart rate is calculated by counting the number of systolic peaks per minute. Figure 6 shows the systolic peaks and peak-to-peak interval of a PPG signal measured by the NeXus-10 MKII device in the dry environment. 

## 4. Results

The PPG signals extracted from the smartphone and the NeXus device for each environment were analyzed in terms of signal quality (signal amplitude and SNR). To eliminate the effect of temperatures and only focus on medium effect, we fixed the temperature at 18 °C. The experimental environment was an indoor environment with regular ambient light condition and air-conditioned room temperature maintained at 18 °C. PPG signal examples acquired by the smartphone in the dry and underwater environments are shown in Figure 7a,b, respectively while a reference PPG signal example acquired by the NeXus device is shown in Figure 7c. The reference PPG signal is always measured in dry environment for both dry and underwater environments since it is required to be gold-standard. As shown in Figure 7a,b, the wet environment changed smartphone PPG signals in terms of shape as well as peak-to-peak interval (or heart rate).

The smartphone application has artifact correction algorithm that operates before giving an estimation of the heart rate [31]. Whenever the signal is affected by any type of artifact, the measurement time increases until the application gets the usable signal. Therefore, the moment when the smartphone shows the final estimated heart rate value is ≈10 s in the dry environment but it increases to ≈20 s in the underwater environment as shown in Table 1.

Table 2 shows signal amplitude, SNR, and heart rate values derived from the PPG signals of each volunteer in dry and underwater environments. As shown in Table 2, average SNR decreases from 12.84 into 4.96 on average and signal amplitude decreases from 0.4 to 0.2 on average. Moreover, heart rate acquired by the smartphone in the dry environment is shown to have 0.05 bpm difference on average compared to the gold-standard while the average difference is 4.8 bpm in underwater environment.

Comparison of measurements of the two environments from the smartphone with the NeXus as a reference device was performed using the two-way repeated measures ANOVA using SPSS (IBM Corp. Released 2017. IBM SPSS Statistics for Windows, Version 25.0. Armonk, NY: IBM Corp). The results revealed that there was a significant interaction effect between Device and Environment (F = 69.6, *p* < 0.05). The interaction effect indicates that the heart rates from smartphone (M = 80.52) were significantly higher than those from NeXus (M = 76.39) in the underwater environment (Mean Difference = 4.13, SE = 1.84, *p* < 0.05) while the heart rates from the smartphone (M = 76.35) and NeXus (M = 75.70) were not statistically different in the dry environment (Mean Difference = 0.652, SE = 0.59, *p* = 0.29).

Figure 8 shows the plot of each measurement’s amplitude for all participants between the dry and underwater environments. A paired-samples *t*-test was conducted to compare the measurement’s amplitude of heart rates in the dry and underwater environments. There was a significant difference in amplitude for the dry environment (M = 0.37, SD = 0.04) and the underwater environment (M = 0.18, SD = 0.04); t (22) = 17.9, *p* < 0.001.

The Bland–Altman plots in Figure 9 show the agreement between the heart rates obtained from the smartphone and NeXus in dry and underwater conditions. Figure 9a shows a bias between the mean differences of 0.65 in heart rates of smartphone and NeXus in dry condition with 95% limits of agreement interval of the mean differences. On the other hand, Figure 9b shows a bias between the mean difference of 4.13 from smartphone in underwater and NeXus in dry condition, as well as an agreement interval with 95% of the mean differences. This implies that the bias between heart rates with smartphone and NeXus are larger in underwater condition than in dry condition.

To evaluate the effect of temperature on the PPG signal measurement in underwater environment, we measured signal amplitude and heart rate accuracy at 45 °C, 18 °C, and 5 °C. Here, the heart rate accuracy is calculated by *Accuracy* (%) = 100% − *PE* where *PE* is the percentage of error and is derived by PE=|measured value−actual value|/actual value×100%. Figure 10 shows the amplitude and heart rate accuracy of the measured PPG signal for varying temperature in underwater and dry environments. The signal amplitude in the underwater environment was on average 32%, 18%, and 14% lower at 45 °C, 18 °C, and 5 °C, respectively, compared those in the dry environment. The accuracy in the underwater environment was on average 5%, 18%, and 14% lower at 45 °C, 18 °C, and 5 °C, respectively, compared those in the dry environment.

## 5. Discussion

There have been studies investigating the effect of different conditions and factors including motion noise artifact, skin tone, nail polishes, age, and ambient light on PPG signal amplitude [8,9,10,11]. For heart rate measurements in underwater condition, several approaches have been proposed [10,11]. Schipke et al. found that 5 min immersions for divers in 6 °C cold ocean water resulted in 10% decrease in SpO_2_ measurement accuracy and 40% decrease in heart rate measurement accuracy. Reyes et al. developed a compound of carbon black powder and polydimethylsiloxane (CB/PDMS) and tested their electrode for underwater ECG measurements which had an average of 6 bpm deviation in the heart rate measurement while in our underwater PPG measurement the average heart rate deviation was 5 bpm. Their methods use ECG electrodes for heart rate measurements, and it remains to be seen if their accuracy remains in salty and non-salty water conditions. Moreover, these studies provide good estimates of heart rate measurements in cold water while the effect of high temperatures (higher than 6 °C) on heart rate measurement was not studied in [21,23].

Khan et al. studied the effects of cold and warm temperatures (5 °C and 45 °C) on PPG signal amplitude and heart rate measurements and concluded that temperature drop from 45 °C to 5 °C decreases the PPG signal from 0.4 V to 0.1 V [12]. In particular, our experiment found that decreasing the water temperature decreases the performance. Moreover, Khan’s study was only conducted in dry conditions and they did not study the effect of different mediums (e.g., water) on the PPG signal amplitude. Our results indicate that changing the medium from air to water and decreasing the temperature from 45 °C to 5 °C decreased the signal amplitude from 0.561 to 0.091. To the authors’ knowledge, no research has studied the effects of different mediums (e.g., water) on PPG sensors signals amplitude and SNR.

As it was expected, our results indicate that there was a significant interaction effect between device and environment (F = 69.6, *p* < 0.05). The interaction effect indicates that the heart rates from dry environment (M = 80.52) were significantly higher than those from underwater environment (M = 76.39) (mean difference = 4.13, SE = 1.84, *p* < 0.05). Our experimental result indicates that PPG measurement gives acceptable accuracy in normal water pressure, normal light condition and temperature maintained at 18 °C. When the temperature decreases, however, a noise resilient peak detection is needed to acquire reliable heart rate information while it is not in the dry environment.

## 6. Conclusions

In this study, we investigated the effects of underwater and dry environments on signal amplitude and SNR values of PPG signals. Experimental results showed that underwater environment impacts the signal amplitude significantly (*p* < 0.05), decreases the average amplitude value by 22% and the average accuracy of the heart rate PPG measurement decreases by 7% on average. The underwater temperature also impacts the heart rate measurement. Decreasing the underwater temperature from 45 °C to 5 °C, increases the difference between the accuracy of underwater and dry environment from 5% to 14% while acceptable accuracy of heart rate estimate provided by FDA approved-device is ±10% of value, not to exceed 5 bpm [41]. Experimental results show that heart rate can be acquired in underwater environment with ≈80% accuracy on average compared to dry environment. Moreover, in underwater condition, experimental results show that temperature significantly affects the accuracy of heart rate estimation as well as signal amplitude. As the temperature decreases from 45 °C to 5 °C, the signal amplitude is shown to decrease by 12 dB. The average temperature of the ocean surface in Texas coasts (Gulf of Mexico) is about 17 °C and it gets as low as 4 °C in January and as high as 24 °C in August according to the national center for environmental information (NOAA) [42]. According to these information and our experimental results, for example, the heart rate could be measured with an average accuracy of 82% in August (at 24 °C) and 60% in January (at 4 °C) in sea water at Texas coasts (Gulf of Mexico), and 84% average accuracy in swimming pools (water temperature around 26 °C).

## Figures and Tables

**Figure 1 sensors-19-02846-f001:**
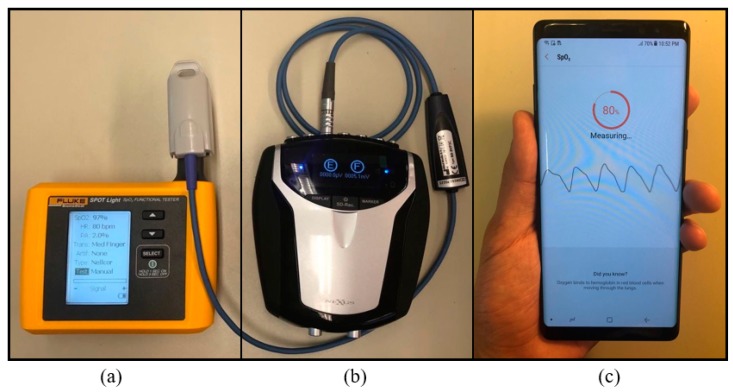
Our measurement and calibration devices used for data acquisition: (**a**) FLUKE device which is an emulation device to calibrate other measurement devices, (**b**) NeXus-10 MKII is FDA approved physiological signal measurement device which we use as our gold standard device in this paper, and (**c**) Samsung Galaxy Note8, a water-resistant device which is capable of measuring PPG in both dry and underwater environments.

**Figure 2 sensors-19-02846-f002:**
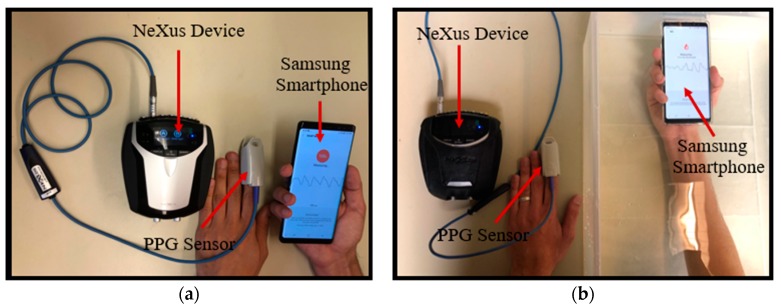
Data acquisition in dry and underwater environments. (**a**) Dry environment: the Samsung Galaxy Note8 PPG sensor is covered with the index finger of the right hand while the PPG sensor of the NeXus-10 MKII device encloses the index finger of the left hand; and (**b**) underwater environment: the Samsung Galaxy Note8 PPG sensor is covered with the index finger of the right hand while the hand is immersed in the transparent bucket filled with water. The PPG sensor of the NeXus-10 MKII device encloses to the index finger of the left hand.

**Figure 3 sensors-19-02846-f003:**
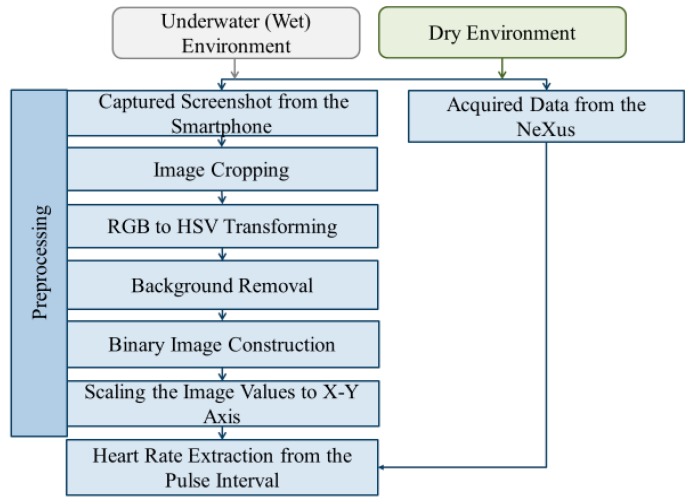
The flowchart of our proposed data processing procedure which consists of acquiring screenshot during the measurement, cropping acquired images, converting RGB image to HSV color space, obtaining a binary image after denoising background from HSV images, and scaling a binary image to values in x-y axis. The heart rate is extracted from peak-to-peak (or pulse) interval from the constructed PPG signal (left). The acquired smartphone PPG signal is compared to gold-standard PPG signal acquired by NeXus-10 MKII (right).

**Figure 4 sensors-19-02846-f004:**
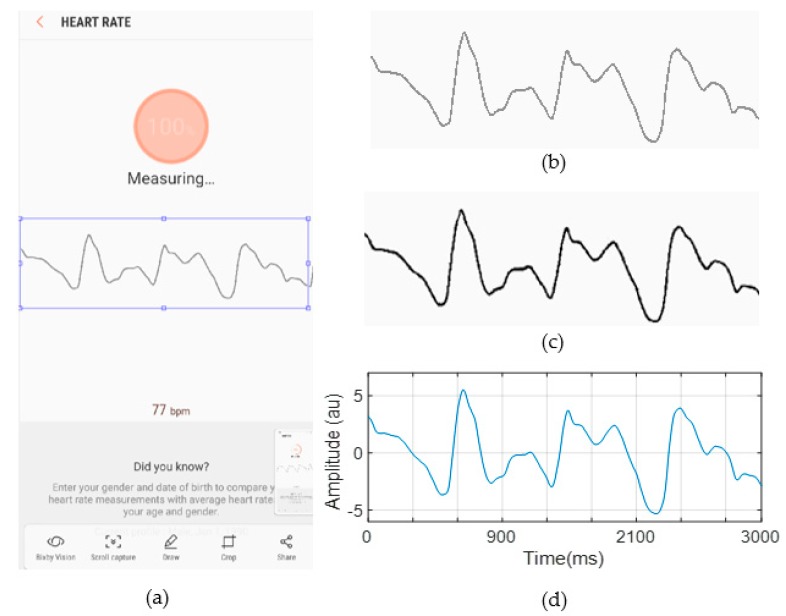
An example of output result at each step of our proposed data processing procedure. (**a**) Screenshot from the smartphone built-in application, (**b**) its cropped image, (**c**) the binary image derived by the RGB to HSV transformation and background removal, and (**d**) extracted signal by scaling the binary image to the *x-y* axis.

**Figure 5 sensors-19-02846-f005:**
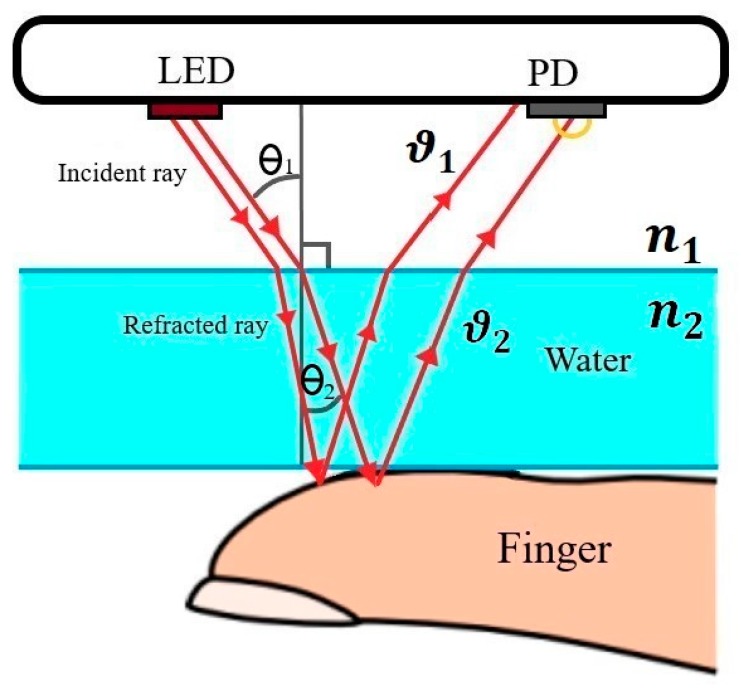
Refraction effect between water and air with the smartphone configuration with an LED emitter and the photodiode (PD) receiver sensor.

**Figure 6 sensors-19-02846-f006:**
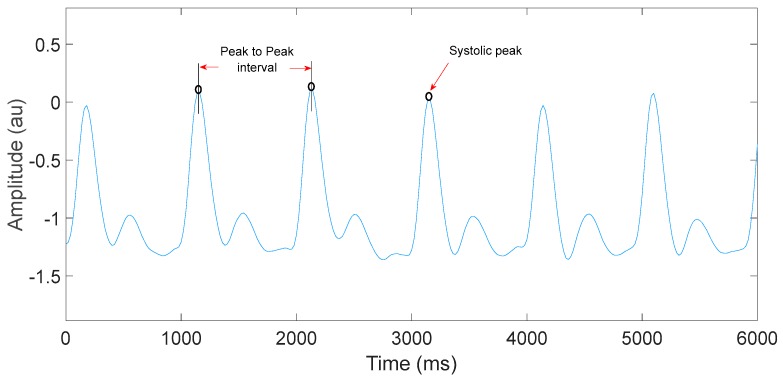
Systolic peaks and peak-to-peak intervals of a PPG signal measured by the NeXus-10 MKII device signal in the dry environment.

**Figure 7 sensors-19-02846-f007:**
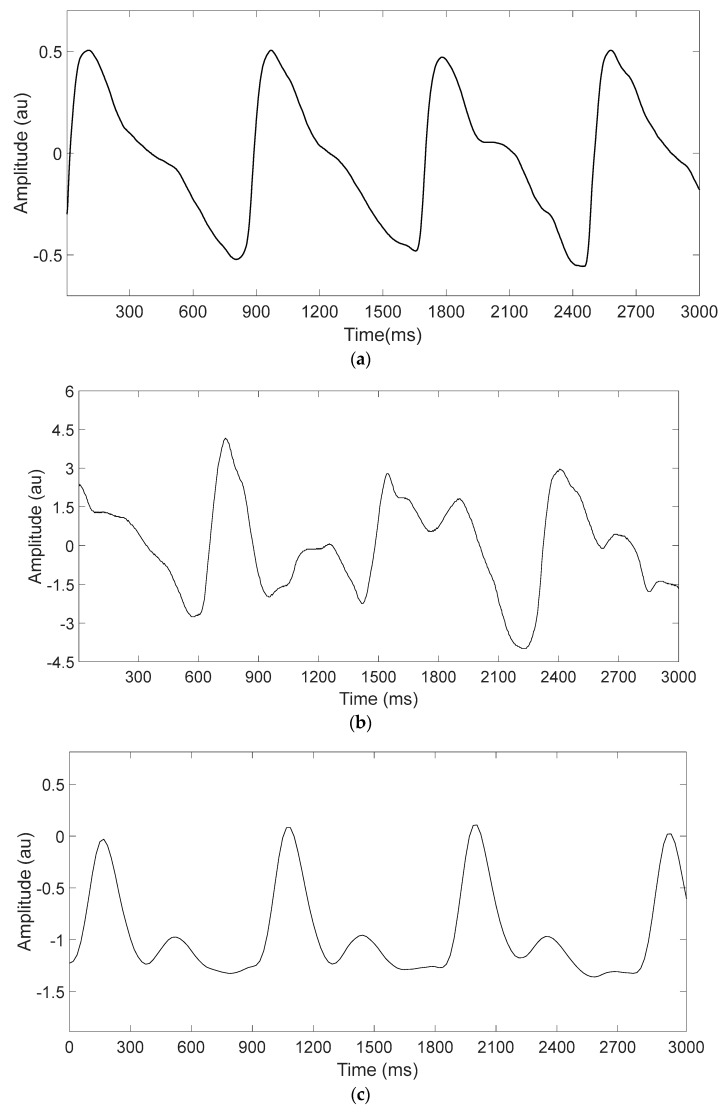
Representative PPG signal examples measured in dry and underwater environments. (**a**) PPG signals measured in dry environment and (**b**) in underwater environment using Samsung Galaxy Note8, and (**c**) a reference PPG signal measured using NeXus-10 MKII.

**Figure 8 sensors-19-02846-f008:**
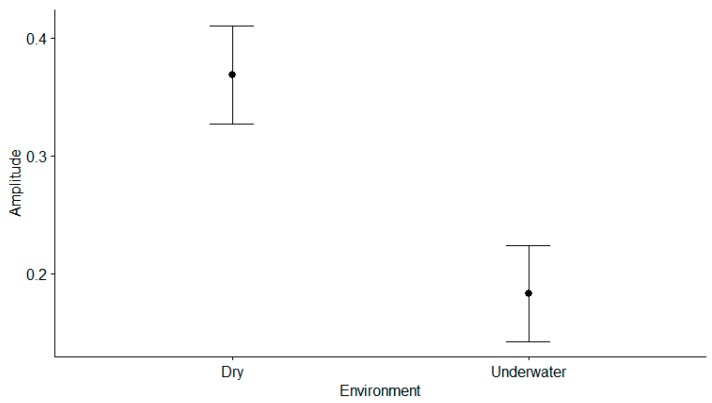
Amplitude plot: left plot shows the amplitude of the dry environment from NeXus-10 MKII and right plot shows the amplitude of underwater environment from Samsung Galaxy Note8.

**Figure 9 sensors-19-02846-f009:**
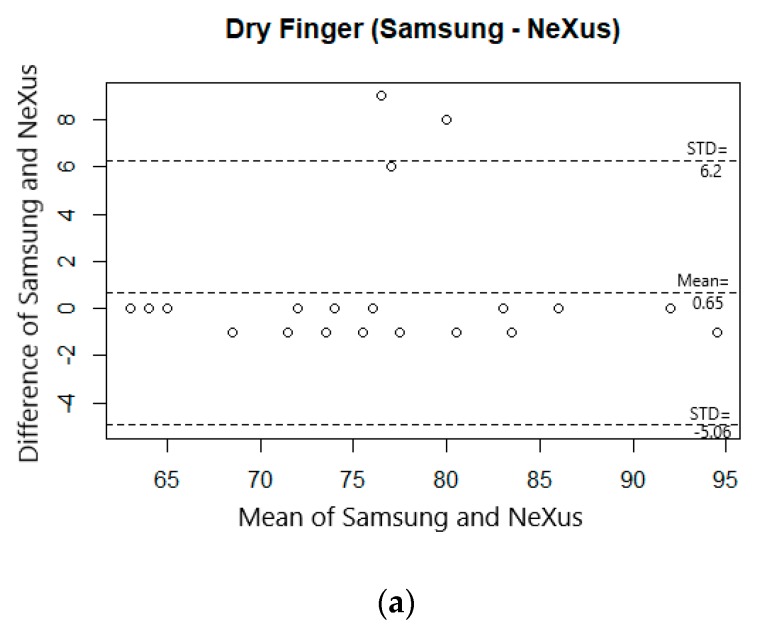
Bland–Altman plots of heart rate measurements after signal processing. (**a**) Heart rate from Samsung Galaxy Note8 and NeXus-10 MKII in dry environment, and (**b**) Heart rate from NeXus-10 MKII in dry and Samsung Galaxy Note8 in wet environment.

**Figure 10 sensors-19-02846-f010:**
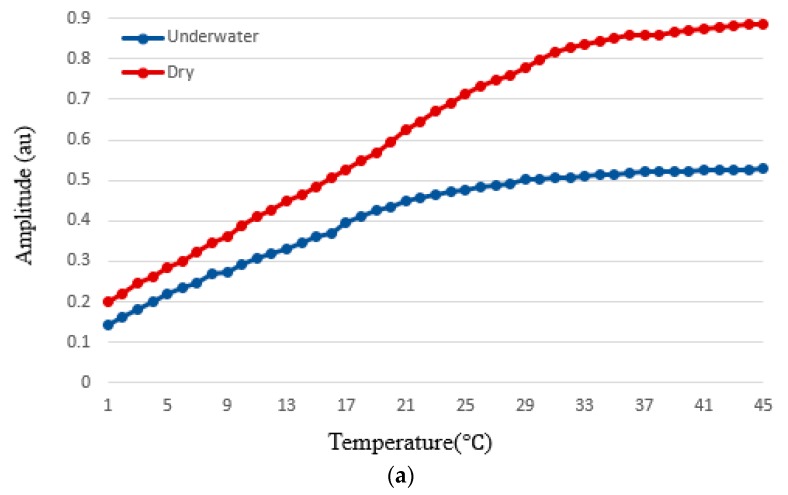
Signal amplitude and heart rate accuracy acquired from smartphone PPG signals for varying temperature in dry (red) and underwater (blue) environments. (**a**) signal amplitude, and (**b**) heart rate accuracy.

**Table 1 sensors-19-02846-t001:** Mean and standard deviation for the measurement duration in dry and underwater environments.

	Dry Environment	Underwater Environment
	Mean ± STD	Mean ± STD
Time (s)	10 ± 2.7	20 ± 3.4

**Table 2 sensors-19-02846-t002:** Heart rate estimation, amplitude and SNR values from the NeXus-10 MKII device and Samsung Galaxy Note8 for dry and underwater environments at baseline temperature (18 °C).

Volunteer	Dry Environment				Underwater Environment				
	Gender & Age	NeXus Heart Rate (bpm)	Smartphone Heart Rate (bpm)	Amplitude Value (au)	SNR (dB)	NeXus Heart Rate (bpm)	Samsung Heart Rate (bpm)	Amplitude Value (au)	SNR (dB)
1	F 30	83	84	0.358	12.5	94	100	0.238	4.3
2	M 32	76	76	0.356	12.9	77	84	0.204	4.2
3	M 23	72	72	0.352	12.1	70	74	0.189	4.1
4	F 37	95	94	0.322	11.8	88	92	0.184	5.2
5	M 26	83	83	0.428	12.8	92	85	0.236	5.5
6	M 29	71	72	0.383	12.6	68	73	0.241	5.3
7	F 45	64	64	0.386	11.7	70	66	0.226	5.6
8	M 27	77	78	0.327	12.1	70	73	0.177	4.2
9	M 33	76	76	0.405	13.2	75	71	0.238	4.4
10	M 26	63	63	0.323	13.1	64	68	0.178	4.2
11	M 38	80	81	0.392	12.8	77	77	0.192	4.4
12	M 25	63	63	0.391	13.8	69	65	0.187	4.6
13	M 27	73	74	0.411	11.4	74	68	0.164	4.3
14	F 72	92	92	0.315	14.1	96	91	0.168	4.8
15	F 19	76	76	0.364	14.3	78	84	0.192	5.5
16	F 21	65	65	0.292	14.5	63	69	0.115	5.1
17	F 23	75	76	0.341	13.2	76	72	0.205	4.2
18	F 22	74	74	0.313	12.9	72	80	0.215	4.4
19	M 20	68	69	0.373	13.2	70	77	0.178	4.2
20	M 19	86	86	0.364	13.1	85	94	0.113	4.5
Mean ± STD		75.9 ± 9	75.95 ± 8.9	0.39 ± 0.14	12.8 ± 1.4	75.4 ± 9	80.2 ± 10.2	0.2 ± 0.1	4.9 ± 0.7

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
