# Peer review of "Monitoring of Heart Rate from Photoplethysmographic Signals Using a Samsung Galaxy Note8 in Underwater Environments"

_sensors, 2019, doi:10.3390/s19132846_

Reviewer 1 Report

Please, see attached document

Author Response

Reviewer: Reviewer 1

We would like to thank the reviewer for the positive judgment of our work and for the remarks, which have led to the improvement of our paper. Below we include a point-by-point reply to the comments. We have modified our paper following these comments and we hope that it is now suitable for publication. The authors have clarified the points as the reviewer suggested and revised the manuscript. Please note that all changes to the text are highlighted in yellow.

General Comments & Questions:

In this study, the authors performed a series of experiments aimed to evaluate the performance of smartphone acquired PPG signals for the task of heart rate estimation in dry and underwater environments and different water temperatures. The smartphone estimates were compared versus an FDA approved PPG device taken as gold reference. Authors reported, as expected, that SNR and amplitude of PPG signals is decreased underwater and at lower temperatures, but heart rate can still be estimated with an increased error of 5 beats-per-minute when compared to estimates in dry conditions. Besides the obtained results, my recommendation is to require them some modifications and clarifications that improve the quality of the paper before it is published.

RESPONSE: Thank you very much for your astute comments. The authors have modified the draft by clarifying/addressing your comments as follows:

Specific Comments & Questions:

(1) Why authors did not perform SpO2 analysis/comparison given that both devices provide that measurement already?

RESPONSE: Thank you very much for your astute comment. As you suggested, the authors have analyzed and compared SpO2 values provided by smartphone (Samsung Galaxy Note 8) and reference (NeXus 10 MK-II) devices as shown in the following Table 1. We compared the mean and standard deviation of the SpO2 measurement errors between smartphone and NeXus devices in dry and wet environments. As a result, the mean and standard deviation are 0.21±0.27% in the dry environment while 1.7±0.63% in the wet environment, which shows that wet environment gives higher error than the dry environment.

Table 1. SpO2 values in dry an underwater environment from Samsung Galaxy Note 8 and NeXus 10 MK-II and the comparison between Samsung and NeXus device.

Volunteer

Samsung Spo2 Wet (%)

NeXus Spo2 Wet (%)

Samsung 

Spo2 Dry (%)

NeXus Spo2 Dry (%)

1       

98

98

98

98

2       

97

100

99

100

3       

97

98

98

98

4       

98

97

97

97

5       

100

98

98

98

6       

98

99

99

99

7       

97

97

97

97

8       

99

98

98

98

9       

100

97

99

98

10     

96

99

100

100

11     

98

97

98

99

12     

95

98

97

99

13     

96

98

98

98

14     

97

97

97

97

15     

98

97

97

97

16     

98

98

98

98

17     

97

96

96

96

18     

96

98

98

99

19     

97

98

98

98

20     

98

96

97

98

21     

99

97

98

98

22     

96

98

97

97

23     

98

99

99

98

To derive SpO2 values from smartphone PPG signals not by built-in software but by ourselves, we need the following calculation using Eqns. (1) and (2), which need the information about AC and DC components of red and infrared smartphone PPG segments. Samsung Galaxy Note 8 only provides a red PPG signal on the screen as shown in Figure 1 below. Hence, the authors could not calculate SpO2 by equations (1) and (2) due to the lack of infrared signal, and did not include SpO2 comparisons in the original draft. 

Figure 1. Screen shot from SpO2 measurement of the Samsung Galaxy Note 8.

(2) In abstract, authors state that results indicate that PPG measurement could be acceptable for underwater activities (swimming, diving, etc.). However, these activities involve conditions not explored in this study, e.g. tons of body movements while swimming and much higher pressure and water depths while diving.

RESPONSE: Thank you for your astute comment. As you pointed out, the underwater activities which you pointed out involve the conditions (e.g. motion noise artifact) which have not been explored in this study.  The authors of this paper have published papers on managing motion noise artifact. Hence, the authors will extend the research with diverse types of motion and noise artifacts. To avoid confusion, the authors have changed the sentence “Experimental results show that heart rate is estimated within 5.06 bpm deviation at 18°Cwhile estimation accuracy decreases as temperature goes down.” as follows:

 -----
[Lines 29-30, Page 1, Section Abstract]: Experimental results show that heart rate is estimated within 5.06 bpm deviation at 18°Cwhile estimation accuracy decreases as temperature goes down.

-----

(3) Introduction, lines 80 to 107. This paragraph combines methodology, results and conclusions. In my opinion, it could be summarized to focus on main contributions of the paper and left the remaining portion for the main body of the paper.

RESPONSE: Thank you very much for your astute comment. As you suggested, the paragraph in lines 80 to 107 has been modified in a way of summarizing the main contribution of the paper. As a result, Lines 80–84 and line 99 are modified as follows in the revised draft, and the authors have move Lines 85-98 to Method section and Lines 100-107 to Conclusion section in the revised draft after modifying them. 

-----

Modified Summary in Introduction Section[Lines 85-90, Page 2, Section Introduction]: In this paper, sensor in the Samsung Galaxy Note 8 in dry and underwater environments. As performance metrics, we consider the accuracy of the heart measurements, signal quality, signal amplitude and signal to noise ratio. We compared acquired information from PPG signals to the reference signal to get and evaluate the performance metrics in dry and wet environments. Here, the heart rate obtained from NeXus-10 MKⅡ device which is FDA-approved is used as a gold standard.

Modified Sentences moved to Method Section[Lines 168-180, Page 5, Section Method]: We extract raw PPG signal from the image of PPG signal on the smartphone screen by our proposed image and signal processing algorithms. Specifically, PPG signal extraction from smartphone starts with image cropping from recorded videos of smartphone screen. Firstly, the recorded RGB images are converted into HSV (Hue, Saturation, Value) color space images. After extracting HSV values from the converted HSV images, background and noise are reduced by the color mask technique in our proposed method. Next, the denoised image is converted to a binary image and the raw PPG signal is finally acquired. Processing raw signal starts with measuring the amplitude of the signal, and the signal to noise ratio (SNR), and the root mean square (RMS) value. In our proposed method, we use systolic peaks, the maximum amplitude value to calculate the heart rate. We evaluate the performance of underwater PPG signal acquisition with our proposed method by applying it to 20 volunteers. We measured PPG signal from 20 volunteers in dry and wet environments. In addition to the effect of different medium, we evaluate the effects of temperature on heart rate measurement accuracy and signal quality.

Modified Sentences moved to Conclusion Section[Lines 393-403, Page 15, Section Conclusion]:Experimental results show that heart rate can be acquired in underwater environment with similar performance compared to dry environment. In underwater condition, results show that temperature significantly affect the accuracy of heart rate estimation as well as signal amplitude. As the temperature decreases, for example from 50°C to 5°C, the signal amplitude decreases by 12 dB. The average temperature of the ocean surface in Texas coasts (Gulf of Mexico) is about 17°C and it gets as low as 4°C in January and as high as 24°C in August according to the national center for environmental information (NOAA)[1]. According to the NOAA information and our calculations, the heart rate could be measured with an accuracy of 82% in August and an accuracy of 60% in January. The average swimming pool water temperature is around 26°C which means an underwater PPG measurement device can measure heart rate with an accuracy of 84%.

-----

(4) In Measurements devices and data acquisition section, authors are requested to provide more information about the devices used. For example, which wavelengths are they using for HR and SpO2 estimation? Are they using the reflection or transmission PPG modalities? Also, provide information about the manufacturer for each device.

RESPONSE: We apologize for this confusion. We use Samsung Galaxy note 8 for data acquisition and Nexus-10 MK-II for reference in this paper. Samsung Galaxy Note 8 is a water and dust resistant smartphone which has a PPG sensor on the rear side for the purpose of measuring heart rate and SpO2 [1]. The NeXuS-10 MK-II is an FDA-approved device made by MINDMEDIA equipped with sensors measuring physiological signals including EEG, ECG, PPG, SpO2, respiration, and EMG [1] uses a reflection mode for the PPG measurements and the NeXus-10 MK-II device uses transmission mode. Both devices use 645 nm Red LED and a 940 nm Infra-Red LEDwavelength for HR and SpO2 estimation. The authors have added this explanation in the revised manuscript as follows:

-----
[Lines 108-114, Page 3, Section Materials]: We use Samsung Galaxy note 8 for data acquisition and Nexus-10 MK-II for reference in this paper. Samsung Galaxy Note 8 is a water and dust resistant smartphone which has a PPG sensor on the rear side for the purpose of measuring heart rate and SpO2 [1]. The NeXuS-10 MK-II is an FDA-approved device made by MINDMEDIA equipped with sensors measuring physiological signals including EEG, ECG, PPG, SpO2, respiration, and EMG [2]. uses a reflection mode for the PPG measurements and the NeXus-10 MK-II device uses transmission mode. Both devices use 645 nm Red LED and a 940 nm Infra-Red LEDwavelength for HR and SpO2 estimation. 

-----

 (5)   It would be really valuable for readers to indicate what is the acceptable accuracy for HR estimates provided by approved FDA devices?

RESPONSE: Thank you very much for your astute comment. The acceptable accuracy of HR estimates provided by Food and Drug Administration (FDA)-approved device is ±10% of value and must not exceed 5 bpm [3]. As the reviewer suggested, the authors have added this information as follows:  

-----
[Lines 392-393, Page 15, Section Conclusion]: Acceptable accuracy of HR estimate provided by FDA approved- device is ±10% of value, not exceed 5 bpm [3].

-----

(6) In Study protocol section, could authors include more information about the studied sample? e.g. mean +- std of height, weight and age, number of males and female volunteers 

RESPONSE: Thank you very much for your astute comment. As you suggested, the authors have added age and gender in Table 3 of the original draft (Table 2 of the revised manuscript).Unfortunately, our approved Institutional Review Board (IRB) (IRB#: IRB2018-688) does not include the procedure of asking weight and height information of the volunteers. Table 2 (see below) in the revised draft has the corresponding information.

-----
[Table 2, Page 11, Section Results]:

Table 2. Heart rate estimation, amplitude and SNR values from the NeXus-10 MK II device and Samsung Galaxy note 8 for dry and underwater environments at baseline temperature (18.

Dry Environment

Underwater Environment

Volunteer

 Gender

&Age

NeXus

Heart Rate

(bpm)

Smartphone

Heart Rate

(bpm)

Amplitude Value

(au)

SNR

(dB)

NeXus

Heart Rate

(bpm)

Samsung Heart Rate

(bpm)

Amplitude

Value

(au)

SNR

(dB)

1      

F 30

83

84

0.358

12.5

94

100

0.238

4.3

2       

M 32

76

76

0.356

12.9

77

84

0.204

4.2

3       

M 23

72

72

0.352

12.1

70

74

0.189

4.1

4       

F 37

95

94

0.322

11.8

88

92

0.184

5.2

5       

M 26

83

83

0.428

12.8

92

85

0.236

5.5

6       

M 29

71

72

0.383

12.6

68

73

0.241

5.3

7       

F 45

64

64

0.386

11.7

70

66

0.226

5.6

8       

M 27

77

78

0.327

12.1

70

73

0.177

4.2

9       

M 33

76

76

0.405

13.2

75

71

0.238

4.4

10     

M 26

63

63

0.323

13.1

64

68

0.178

4.2

11     

M 38

80

81

0.392

12.8

77

77

0.192

4.4

12     

M 25

63

63

0.391

13.8

69

65

0.187

4.6

13     

M 27

73

74

0.411

11.4

74

68

0.164

4.3

14     

F 72

92

92

0.315

14.1

96

91

0.168

4.8

15     

F 19

76

76

0.364

14.3

78

84

0.192

5.5

16     

F 21

65

65

0.292

14.5

63

69

0.115

5.1

17     

F 23

75

76

0.341

13.2

76

72

0.205

4.2

18     

F 22

74

74

0.313

12.9

72

80

0.215

4.4

19     

M 20

68

69

0.373

13.2

70

77

0.178

4.2

20     

M 19

86

86

0.364

13.1

85

94

0.113

4.5

Mean±STD

75.9±9

75.95±8.9

0.39±0.14

12.8 ± 1.4

75.4±9

80.2±10.2

0.2±0.1

4.9± 0.7

-----

(7) In Study protocol section, was the hand selection randomized or the same for all subjects? 

RESPONSE: We apologize for this confusion. In our study, the index finger of left hand was connected to the NeXus-10 MK-IIdevice and the index finger of the right hand was connected to the Samsung Galaxy smartphone. This hand selection was not randomized but the same for all subjects. The authors have added the following explanation in the revised manuscript as follows:

----

[Lines 125-127, Page 3, Section Materials]:In our study, the index finger of left hand was connected to the NeXus-10 MK-IIdevice and the index finger of the right hand was connected to the Samsung Galaxy smartphone. This hand selection was not randomized but the same for all volunteers.

----

 (8) In Study protocol section, which type of calibration was performed? Why it was performed after taking the measurements and not before?

RESPONSE: We apologize for this confusion.The calibration process was done on each volunteer before each data acquisition and the authors have used SPOT light pulse oximeter PPG analyzer [6] (manufactured by FLUKE Biomedical) which generates artificial PPG signals (see Figure 3b).  This calibration is performed to validate that smartphones and NeXus devices are working correctly before the actual measurement. The authors have added this in the revised draft as follows:

 ----

[Lines 135-138, Page 4, Section Materials]:The calibration process was done on each volunteer before each data acquisition and the authors have used SPOT light pulse oximeter analyzer [6] (manufactured by FLUKE Biomedical) which generates artificial PPG signals. This calibration is performed to validate that smartphones and NeXus devices are working correctly before the actual measurement. 

----

(a)

(b)

(c)

Figure2.(a) Samsung Smartphone attached to the FLUKE device during calibration process, (b) Output PPG signal results of FLUKE SPOT Light functional tester, (c) Output calibrated signal results from Samsung Galaxy Note 8

(9)    In Study protocol section, a. Dry environment. First idea was already said in previous paragraph.

RESPONSE: We apologize for this confusion. The authors have deleted the repeated sentences (lines 152-154 of the original draft) in the revised draft.

(10)   In Study protocol section, b. Underwater environment. Was there a time left in between consecutive temperature conditions? While only the hand was immersed? It would be interesting if the whole body was immersed as it elicits different responses to cardiorespiratory system via the nervous system.

RESPONSE: Thank you for your precious comment. We add ice to cool the water down. Usually, it took 40 seconds for the temperature of water goes down by 1℃. Hence, we first measure data at 50℃. Then, the data at 18℃ is measured 21 minutes 20 seconds after we measure at 50℃. After 6 minutes and 52 seconds, the data is measured at 5℃. Currently, our IRB is confined to ask subjects to immerse their hands only but the authors will extend the study to make the whole body to be immersed in future as the reviewer suggested. We added this explanation in the revised draft.

----

[Lines 147-151, Page 4, Section Materials]: For different temperature data acquisition procedure the water was first heated up to 50℃ and the volunteer was asked to immerse their finger in the warm water. We add ice to cool the water down. Usually, it took 40 seconds for the temperature of water goes down by 1℃. Hence, we first measure data at 50℃. Then, the data at 18℃ is measured 21 minutes 20 seconds after we measure at 50℃. After 6 minutes and 52 seconds, the data is measured at 5℃.

 ----

(11)  In Data processing section, about the signal extracted from PPG device. How can you ensure that the displayed signal corresponds to the raw PPG data and not to a modified signal for display purposes as it is often the case PPG devices? I´ve been using the Note 8 SpO2 app and it appears that it automatically normalizes the amplitude of the signal displayed

RESPONSE: Thank you for the astute comment. We performed movement test with high motion, and we found that the Samsung’s displayed PPG signal has a saturation point and it didn’t normalize the overshooting values. The smartphone simply cuts off the maximum and minimum values when the amplitude exceeds the saturation value as shown in below figure. As shown in below figure when certain noise occurs (e.g., motion noise artifact, underwater environment) the smartphone displays a saturated maximum and minimum cap value which indicates that the displayed values are not normalized. The saturated points of PPG signal from the Smartphone are shown by red box in Figure 6b.

(a)                                                                         (b)

Figure 3. Displayed signal in noise present :(a) Original raw saturated signal (b) Indication of maximum and minimum saturation (blue line) saturated parts (red box).

(12)  In Preprocessing and heart rate extraction section. Regarding the screenshots, how were they taken? Did that procedure introduce extra motion artifact movements to the acquisition? or they were activated remotely? How were the different and consecutive screenshots aligned?

RESPONSE:We apologize for this confusion. The screenshots were taken by the Mobizen screen recording smartphone app. The screen recording application provides full HD recording at 60 fps. Setup and control is done by the experimenter. Since the Mobizen app is built-in smartphone app, it does not introduce extra motion artifact movements to the acquisition. Different and consecutive screenshots are aligned by a stitching  technique [2, 3]. The authors have added the explanation in the revised manuscript as follows:

-----

[Lines 186-189, Page 5, Section Materials]: The screenshots were taken by the Mobizen screen recording application. The screen recording application provides full HD recording at 60 fps. The volunteer was instructed to the action in a matter that no extra motion artifact movement will be added. We used a stitching technique to align consecutive screen shots after the screenshots were acquired [2, 3].

-----

(13)  In Preprocessing and heart rate extraction section. How were the final amplitude values obtained? from the screenshots? How do you arrive to the scale shown, for example in Fig. 4.d?

RESPONSE: We apologize for this confusion. Yes, we obtain the final amplitude values from the screenshots. Hence, the unit of y-axis in Fig. 4d is arbitrary unit (au). 

(14) In Statistical analysis section, what was considered as noise to compute SNR? Also, for VRMS, did the authors assume that PPG signal was a pure sine wave? Otherwise the RMS values will be different, e.g. for triangle and sawtooth waves it is Am/sqrt(3), for square wave it is Am. My suggestion is to work only with amplitudes (Am).

RESPONSE: Thank you for your astute comment. We apologize for this confusion. The average noise power  is calculated when only noise is present using the equation where  is the k-th sample of the noise signal n, and Nis the number of samples As you suggested, the authors have changed all the RMS values to amplitude in the revised manuscript.

 (15)  In Table 1, are the data presented the average from all subjects or an example from a single subject?

RESPONSE: We apologize for this confusion. All the values shown in Table 1 are not RMS values but average values. The averaged values of pulse amplitudes over 20 persons for each temperature, device, and condition are listed in Table 1 of the original draft. To make clear about each value, the authors have deleted Table 1 of the original draft and added Table 2 in the revised draft which has all the values from 20 persons as well as their average value.

 (16)  Regarding the PPG amplitudes, to complement the information, authors are encouraged to provide the SNR values for each volunteer as for the other values in Table 3 .

RESPONSE: Thank you very much for your astute comments. The authors have provided the SNR values for all volunteers in Table 3 of the original draft (Table 2 of the revised draft) as follows:

-----

[Table 2, Page 11, Section Results] 

Table 2. Heart rate estimation, amplitude and SNR values from the NeXus-10 MK II device and Samsung Galaxy note 8 for dry and underwater environments at baseline temperature (18.

Dry Environment

Underwater Environment

Volunteer

 Gender

&Age

NeXus

Heart Rate

(bpm)

Smartphone

Heart Rate

(bpm)

Amplitude Value

(au)

SNR

(dB)

NeXus

Heart Rate

(bpm)

Samsung Heart Rate

(bpm)

Amplitude

Value

(au)

SNR

(dB)

1       

F 30

83

84

0.358

12.5

94

100

0.238

4.3

2       

M 32

76

76

0.356

12.9

77

84

0.204

4.2

3       

M 23

72

72

0.352

12.1

70

74

0.189

4.1

4       

F 37

95

94

0.322

11.8

88

92

0.184

5.2

5       

M 26

83

83

0.428

12.8

92

85

0.236

5.5

6       

M 29

71

72

0.383

12.6

68

73

0.241

5.3

7       

F 45

64

64

0.386

11.7

70

66

0.226

5.6

8       

M 27

77

78

0.327

12.1

70

73

0.177

4.2

9       

M 33

76

76

0.405

13.2

75

71

0.238

4.4

10     

M 26

63

63

0.323

13.1

64

68

0.178

4.2

11     

M 38

80

81

0.392

12.8

77

77

0.192

4.4

12     

M 25

63

63

0.391

13.8

69

65

0.187

4.6

13     

M 27

73

74

0.411

11.4

74

68

0.164

4.3

14     

F 72

92

92

0.315

14.1

96

91

0.168

4.8

15     

F 19

76

76

0.364

14.3

78

84

0.192

5.5

16     

F 21

65

65

0.292

14.5

63

69

0.115

5.1

17     

F 23

75

76

0.341

13.2

76

72

0.205

4.2

18     

F 22

74

74

0.313

12.9

72

80

0.215

4.4

19     

M 20

68

69

0.373

13.2

70

77

0.178

4.2

20     

M 19

86

86

0.364

13.1

85

94

0.113

4.5

Mean ± STD

75.9±9

75.95±8.9

0.39±0.14

12.8 ± 1.4

75.4±9

80.2±10.2

0.2±0.1

4.9± 0.7

-----

(17)  Regarding the HR estimates underwater, could the difference versus Nexus be due to the low quality of the signals underwater that the built-in Samsung algorithm in the smartphone does not account for? It should be very interesting to apply other HR estimation algorithms to verify this.

RESPONSE: Thank you very much for your astute comment. The authors did not use Samsung’s built-in algorithm and used our HR estimation method based on peak-to-peak intervals. As you suggested, the authors have compared the HR estimation performance of our algorithm to that of Samsung’s algorithm as shown in Table 2. Here NeXus 10 MK-II device is considered as gold standard. Table 2 shows that the mean and standard deviation of HR estimation errors is 12.7 3.56± and 4.8±3.01 for Samsung’s and our algorithms, respectively. 

(a)

(b)

(c)

Figure 4.Heart rate measurement from underwater PPG signal using different algorithms: (a) Example of a PPG signal from Samsung Galaxy Note 8 in underwater environment from one volunteer,  (b) The results of peak finder and peak to peak interval calculation, and (c) calculated value from the built in Samsung Galaxy Note 8 for Volunteer number 1.

Table 2.Comparsion between Samsung built in App and NeXus and our algorithm

Volunteer

Samsung’s HR Estimation Algorithm

Our Developed HR Estimation

Algorithm

NeXus 

Device

(Gold Standard)

1       

116

100

94

2       

92

84

77

3       

77

74

70

4       

95

92

88

5       

98

85

92

6       

78

73

68

7       

81

66

70

8       

76

73

70

9       

75

71

75

10     

73

68

64

11     

84

77

77

12     

77

65

69

13     

77

68

74

14     

114

91

96

15     

88

84

78

16     

72

69

63

17     

85

72

76

18     

81

80

72

19     

79

77

70

20     

94

94

85

(18)  In Figure 8, does this corresponds to the smartphone or Nexus device? If just one, include the amplitude for the other one. If both, indicate it as the difference

RESPONSE: Weapologize for this confusion. Figure 8 corresponds to amplitudes of dry and wet environments of only smartphone. Since NeXus devices could not be immersed due to their lackness of ingress protection (IP), we could not have amplitude data of NeXus devices in wet environment. 

 [Figure 8, Page 12, Section Results]:

Figure 8.Amplitude plot: Left plot shows the amplitude of the dry environment from NeXus-10 MKⅡ and right plot shows the amplitude of underwater environment from Samsung Galaxy Note 8.

-----

(19)  Regarding the Bland-Altman analysis, as far as I know, Bland-Altman plots the difference versus the average of the two measurements (devices). In the cases presented, it appears like the authors are using only scatter plots for the average HR for each subject. Please, check and perform adequate Bland-Altman analysis between both devices for each condition.

RESPONSE: We apologize for this confusion. Thank you very much for your astute comment. The Bland – Altman plots have been corrected. The new Bland-Altman plots are shown below and replaced with the Bland-Altman plots in the paper.

-----

[Lines 337-344, Pages 13-14, Section Results]: The Bland-Altman plots in figure 9 show the agreement between the heart rates obtained from the smartphone and NeXus in dry and underwater conditions. Figure 9a shows a bias between the mean differences of 0.65 in heart rates of smartphone and NeXus in dry condition with 95% limits of agreement interval of the mean differences. On the other hand, figure 9b shows a bias between the mean difference of 4.13 from smartphone in underwater and NeXus in dry condition, as well as an agreement interval with 95% of the mean differences. This implies that the bias between heart rates with smartphone and NeXus are larger in underwater condition than in dry condition. The difference of Samsung and NeXus comes from Samsung – Nexus.

 [Figure 9, Page 13, Section Results]:

(a)

(b)

Figure 9.Bland-Altman plots of heart rate measurements after signal processing. a) Heart rate from Samsung Galaxy Note 8 and NeXus-10 MKⅡ in dry environment, b) Heart rate from NeXus-10 MKⅡ in dry and Samsung Galaxy Note 8 in wet environment.

 ----

(20)  It would be highly appreciated if authors provided a more detailed discussion about results and contrast their findings with previous studies and what was expected from the current body of knowledge.

RESPONSE: Thank you for your astute comments. As you suggested, the authors have added the following sentences in Discussion of the revised manuscripts

-----

[Lines 359-385, Pages 14-15, Section Discussion]: There are several research studying the effect of different conditions and factors including motion noise artifact, skin tone, nail polishes, age and ambient light on PPG signal quality [8-11]. For heart rate measurements in underwater conditions, several approaches have been proposed in the literature [10-11]. J. Schipke et.alfound that 5 minute immersions for divers in 6°C cold ocean water resulted in 10% decrease in SpO2 measurement accuracy and 40% decrease in heart rate measurement accuracy. Reyeset.aldeveloped a comprised mixture of carbon black powder and poly dimethylsiloxane (CB/PDMS) and tested their electrode for underwater environment ECG measurements which had an average of 6 bpm heart rate measurement deviation while in our underwater PPG measurement the average heart rate deviation was 5 bpm. The mentioned methods use ECG electrodes for heart rate measurements, and it remains to be seen if their accuracy remains in different water conditions (salty and non-salty water). Moreover, these studies provide good estimates of heart rate measurements in cold water, while the effect of higher temperatures on heart rate measurement was not studied.

M. Khan et.alstudied the effects of different temperatures (5°C,45°C) on PPG signal quality and heart rate measurements and concluded that a drop in temperature from 45°C to 5°C decreases the PPG signal from 0.4V to 0.1V (4x decrease) [12]. In particular, we found that increasing the water temperature to 50°C increases the heart rate measurement accuracy to up to 85%. Moreover, the mentioned study was only conducted in dry conditions and they did not study the water effect in different temperatures on the PPG signal quality. Our results indicated that changing the median from air to water and decreasing the temperature from 50°C to 5°C decreased the signal amplitude from 0.561 to 0.091 (6x decrease). to the authors’ knowledge no research has studied the effects of water on PPG sensors signal quality. 

As it was expected our results indicate that there was a significant interaction effect between device and environment (F(1,22) = 69.6, p<.05). The interaction effect indicates that the heart rates from dry environment (M=80.52) were significantly higher than those from underwater environment (M=76.39) (Mean Difference=4.13, SE=1.84, p<.05).< span="">

The current study shows that in a controlled environment with controlled light condition, water pressure, when the temperature is not lower than 18°C, the PPG measurement gives acceptable accuracy. However, when the temperature decreases a noise resilient peak detection is needed to acquire reliable heart rate information, while it’s not needed in the dry environment. 

-----

 Minor Comments & Questions:  

 (21)  Abstract, line 11, check spelling of SpO2

RESPONSE: We apologize for this typo. The authors have corrected “Spo2 “into “SpO2” as follows:

-----
[Line 14, Page 1, Section Abstract] …oxygen saturation (SpO2).

-----

 (22)  Abstract, line 14, I suggest rewrite as “N=20 participants, for the task of heart rate estimation.”

RESPONSE: Thank you. We rewrote it in the revised draft as you suggested as follows:

-----
[Line 17, Page 1, Section Abstract]… N=20 volunteers, for the task of heart rate estimation.

-----

(23)  Abstract, line 16, with a gold standard PPG device (Nexus)

RESPONSE: Thank you for your astute comment. Authors changed the sentence “…with a gold standard device (NeXus device)” to “…with a gold standard PPG device (Nexus-10 MKⅡ) as follows:

-----
[Lines 19, Page 1, Section Abstract]…with a gold standard PPG device (NeXus-10 MKⅡ).

-----

(24)  Introduction, line 92, please check if dd to noise ratio is correct or it should say signal to noise 3 ratio (SNR)

RESPONSE: We apologize for this confusionThe authors have fixed this typo in the revised manuscript. “dd” has been changed to “signal” as follows:

-----
[Line 175, Page 5, Section Method]: …amplitude of the signal, and the signal to noise ratio (SNR).

-----

 (25)  My suggestion is to use the word volunteers instead of participants throughout the document.

RESPONSE: Thank you for your astute comments. The authors have replaced the word “participants” with by “volunteers” throughout the revised manuscript.

(26)  In Study protocol section, line 132 I think it should say 18 to 80 years

RESPONSE: Thank you for your astute comments. As you suggested, the authors have changed “18 to 80”into “18 to 80 years” in the revised manuscript as follows:

-----
[Line 125, Page 3, Section Materials]:…the range of 18 to 80 years.

-----

 (27)  In Study protocol section, b. Underwater environment, authors state that light and temperature condition were mentioned in subsection (a) which is not the case done.

RESPONSE: We apologize for this confusion. Since light and temperature condition were not mentioned in subsection (a) as you pointed out, the authors have deleted the sentence “with the same light and temperature condition as mentioned in subsection” of the revised manuscript.

(28)  Figure 2 seems to be truncated at the right side

RESPONSE: Thank you for your astute comment. Figure 2 has been changed and justified in the revised manuscript as follows:

-----
[Figure 2, Page 5, Section Materials]:

(a)

(b)

Figure 2.Dry and underwater environment protocols. a) Dry environment: the Samsung Galaxy Note 8 PPG sensor is covered with the index finger of the right hand and the PPG sensor of the NeXus-10 MKⅡ device is attached to the left hands index finger, b) Underwater environment protocol: the Samsung Galaxy Note 8 PPG sensor is covered with the index finger of the right hand while the hand is immersed in the transparent bucket filled with water. A same time the PPG sensor of the NeXus-10 MK Ⅱ device is attached to the left hands index finger.

 -----

(29)  In Figure 7, I would emphasize that (a) and (b) correspond to smartphone PPG. What about (c) is that from a dry or underwater condition?

RESPONSE: We apologize for this confusion. Figure 7c Is the NeXus device output signal in dry condition. The authors have modified the caption of Figure 7 in the revised manuscript as follows:

-----

[Figure 7, Pages 9-10, Section Results]:

            (a)         (b)

(c)

Figure 7.Representative PPG signals measured in dry and underwater environments. a) PPG signal measured in underwater environment using Samsung Galaxy Note 8, b) PPG signal measured in dry environment using Samsung Galaxy Note 8, and c) PPG signal measured in dry environment using NeXus-10 MK II.

-----

 (30)  In the text (lines 306 and 307) and in the Table title, authors say that “Table 3 shows the results of heart rate estimation from the NeXus signal, smartphone signal, and the smartphone built-in application for all participants and the two-protocol environment.” However, one of the HR estimates obtained from smartphone is missing and the RMS value is presented instead. Another description of Table 3 is provided (lines 312 and 313) and it makes more sense. Please check and correct.

RESPONSE: We apologize for this confusion. The RMS values in Table 3 have been removed and the authors replaced it by amplitude following your comment (14). In the revised draft, the authors listed all the HR estimates. Moreover, as you suggested, we kept the lines 312-313 in the revised draft.

----

[Table 2, Page 11, Section Results]:

Table 2. Heart rate estimation, amplitude and SNR values from the NeXus-10 MK II device and Samsung Galaxy note 8 for dry and underwater environments at baseline temperature (18.

Dry Environment

Underwater Environment

Volunteer

 Gender

&Age

NeXus

Heart Rate

(bpm)

Smartphone

Heart Rate

(bpm)

Amplitude Value

(au)

SNR

(dB)

NeXus

Heart Rate

(bpm)

Samsung Heart Rate

(bpm)

Amplitude

Value

(au)

SNR

(dB)

1       

F 30

83

84

0.358

12.5

94

100

0.238

4.3

2       

M 32

76

76

0.356

12.9

77

84

0.204

4.2

3       

M 23

72

72

0.352

12.1

70

74

0.189

4.1

4       

F 37

95

94

0.322

11.8

88

92

0.184

5.2

5       

M 26

83

83

0.428

12.8

92

85

0.236

5.5

6       

M 29

71

72

0.383

12.6

68

73

0.241

5.3

7       

F 45

64

64

0.386

11.7

70

66

0.226

5.6

8       

M 27

77

78

0.327

12.1

70

73

0.177

4.2

9       

M 33

76

76

0.405

13.2

75

71

0.238

4.4

10     

M 26

63

63

0.323

13.1

64

68

0.178

4.2

11     

M 38

80

81

0.392

12.8

77

77

0.192

4.4

12     

M 25

63

63

0.391

13.8

69

65

0.187

4.6

13     

M 27

73

74

0.411

11.4

74

68

0.164

4.3

14     

F 72

92

92

0.315

14.1

96

91

0.168

4.8

15     

F 19

76

76

0.364

14.3

78

84

0.192

5.5

16     

F 21

65

65

0.292

14.5

63

69

0.115

5.1

17     

F 23

75

76

0.341

13.2

76

72

0.205

4.2

18     

F 22

74

74

0.313

12.9

72

80

0.215

4.4

19     

M 20

68

69

0.373

13.2

70

77

0.178

4.2

20     

M 19

86

86

0.364

13.1

85

94

0.113

4.5

Mean±STD

75.9±9

75.95±8.9

0.39±0.14

12.8 ± 1.4

75.4±9

80.2±10.2

0.2±0.1

4.9± 0.7

 -----

 (31)  In Figure 10, my recommendation is to plot each case (amplitude and accuracy) in a single graph and use different legend labels and line styles/color to differentiate

RESPONSE: Thank you for your astute comment. Figure 10 has been modified. The amplitude graph and accuracy graph has been merged in underwater and dry condition to build a single graph. The different conditions have been color coded (Blue for underwater and Red for dry condition) in the revised manuscript as follows:

-----
[Figure 10, Page 14, Section Results]:

(a)

(b)

Figure10.Accuracy of heart rate measurement and amplitude comparison for different temperatures. a) Underwater amplitude (blue) and dry amplitude (red), b) underwater accuracy (blue) and dry accuracy (red).

-----

 (32)  References section. Please, employ a uniform reference style

RESPONSE: Thank you for your astute comment. The reference style has been unified and justified in the revised manuscript as follows:

----

[Lines 405, Page 15 to Lines 483, Page 17, Section References]:

References

 [1]        WHO, "Cardiovascular diseases (CVDs)."

[2]        W. G. Members et al., "Heart disease and stroke statistics—2017 update: a report from the American Heart Association," Circulation, vol. 135, no. 10, p. e146, 2017.

[3]        D. Mozaffarian et al., "Heart disease and stroke statistics--2015 update: a report from the American Heart Association," (in eng), Circulation, vol. 131, no. 4, pp. e29-322, Jan 27 2015.

[4]        J. W. Chong et al., "Motion and Noise Artifact-Resilient Atrial Fibrillation Detection using a Smartphone," IEEE Journal on Emerging and Selected Topics in Circuits and Systems, 2018.

[5]        M. Sardana et al., "PERFORMANCE AND USABILITY OF A NOVEL SMARTPHONE APPLICATION FOR ATRIAL FIBRILLATION DETECTION IN AN AMBULATORY POPULATION REFERRED FOR CARDIAC MONITORING," Journal of the American College of Cardiology, vol. 67, no. 13 Supplement, p. 844, 2016.

[6]        B. Askarian, F. Tabei, A. Askarian, and J. W. Chong, "An affordable and easy-to-use diagnostic method for keratoconus detection using a smartphone," in Medical Imaging 2018: Computer-Aided Diagnosis, 2018, vol. 10575, p. 1057512: International Society for Optics and Photonics.

[7]        F. Tabei, R. Kumar, T. N. Phan, D. D. McManus, and J. W. Chong, "A Novel Personalized Motion and Noise Artifact (MNA) Detection Method for Smartphone Photoplethysmograph (PPG) Signals," IEEE Access, vol. 6, pp. 60498-60512, 2018.

[8]        A. A. Safavi, A. Keshavarz-Haddad, S. Khoubani, S. Mosharraf-Dehkordi, A. Dehghani-Pilehvarani, and F. S. Tabei, "A remote elderly monitoring system with localizing based on Wireless Sensor Network," in Computer Design and Applications (ICCDA), 2010 International Conference on, 2010, vol. 2, pp. V2-553-V2-557: IEEE.

[9]        O. Bazgir, S. A. H. Habibi, L. Palma, P. Pierleoni, and S. Nafees, "A Classification System for Assessment and Home Monitoring of Tremor in Patients with Parkinson's Disease," Journal of medical signals and sensors, vol. 8, no. 2, p. 65, 2018.

[10]       J. Pietilä et al., "Evaluation of the accuracy and reliability for photoplethysmography based heart rate and beat-to-beat detection during daily activities," in EMBEC & NBC 2017: Springer, 2017, pp. 145-148.

[11]       J. Allen, "Photoplethysmography and its application in clinical physiological measurement," Physiological measurement, vol. 28, no. 3, p. R1, 2007.

[12]       T. Tamura, Y. Maeda, M. Sekine, and M. Yoshida, "Wearable photoplethysmographic sensors—past and present," Electronics, vol. 3, no. 2, pp. 282-302, 2014.

[13]       V. R. Pamula et al., "A 172 µWCompressively Sampled Photoplethysmographic (PPG) Readout ASIC With Heart Rate Estimation Directly From Compressively Sampled Data," IEEE transactions on biomedical circuits and systems, vol. 11, no. 3, pp. 487-496, 2017.

[14]       A. V. J. Challoner, "Photoelectric plethysmography for estimating cutaneous blood flow," Non-invasive physiological measurements, vol. 1, pp. 125-151, 1979.

[15]       D. Phan, L. Y. Siong, P. N. Pathirana, and A. Seneviratne, "Smartwatch: Performance evaluation for long-term heart rate monitoring," in 2015 International Symposium on Bioelectronics and Bioinformatics (ISBB), 2015, pp. 144-147: IEEE.

[16]       R. Wijaya, A. Setijadi, T. L. Mengko, and R. K. Mengko, "Heart rate data collecting using smart watch," in 2014 IEEE 4th International Conference on System Engineering and Technology (ICSET), 2014, vol. 4, pp. 1-3: IEEE.

[17]       M. Elgendi, "On the analysis of fingertip photoplethysmogram signals," Current cardiology reviews, vol. 8, no. 1, pp. 14-25, 2012.

[18]       M. Khan, C. G. Pretty, A. C. Amies, R. Elliott, G. M. Shaw, and J. G. Chase, "Investigating the effects of temperature on photoplethysmography," IFAC-PapersOnLine, vol. 48, no. 20, pp. 360-365, 2015.

[19]       T. Breskovic et al., "Cardiovascular changes during underwater static and dynamic breath-hold dives in trained divers," Journal of applied physiology, vol. 111, no. 3, pp. 673-678, 2011.

[20]       B. A. Reyes et al., "Novel electrodes for underwater ECG monitoring," IEEE Transactions on Biomedical Engineering, vol. 61, no. 6, pp. 1863-1876, 2014.

[21]       W. Gregson et al., "Influence of cold water immersion on limb and cutaneous blood flow at rest," The American journal of sports medicine, vol. 39, no. 6, pp. 1316-1323, 2011.

[22]       J. Schipke and M. Pelzer, "Effect of immersion, submersion, and scuba diving on heart rate variability," British Journal of Sports Medicine, vol. 35, no. 3, pp. 174-180, 2001.

[23]       W. M. Silvers and D. G. Dolny, "Comparison and reproducibility of sEMG during manual muscle testing on land and in water," Journal of electromyography and kinesiology, vol. 21, no. 1, pp. 95-101, 2011.

[24]       A. Rainoldi, C. Cescon, A. Bottin, R. Casale, and I. Caruso, "Surface EMG alterations induced by underwater recording," Journal of Electromyography and Kinesiology, vol. 14, no. 3, pp. 325-331, 2004.

[25]       K. Masumoto, S.-i. Takasugi, N. Hotta, K. Fujishima, and Y. Iwamoto, "A comparison of muscle activity and heart rate response during backward and forward walking on an underwater treadmill," Gait & posture, vol. 25, no. 2, pp. 222-228, 2007.

[26]       N. c. f. e. i. (NOAA).

[27]       R. Delgado-Gonzalo, J. Parak, A. Tarniceriu, P. Renevey, M. Bertschi, and I. Korhonen, "Evaluation of accuracy and reliability of PulseOn optical heart rate monitoring device," in 2015 37th Annual International Conference of the IEEE Engineering in Medicine and Biology Society (EMBC), 2015, pp. 430-433.

[28]       G. E. White, S. G. Rhind, and G. D. Wells, "The effect of various cold-water immersion protocols on exercise-induced inflammatory response and functional recovery from high-intensity sprint exercise," European journal of applied physiology, vol. 114, no. 11, pp. 2353-2367, 2014.

----

-------------------------------------------------------------------------------------------

Added References

 [1] N. c. f. e. i. (NOAA).

[2]S. Mann and R. W. Picard, "Virtual bellows: Constructing high quality stills from video," in Proceedings of 1st International Conference on Image Processing, 1994, vol. 1, pp. 363-367: IEEE.[3]        R. Szeliski, "Image alignment and stitching: A tutorial," Foundations and Trends® in Computer Graphics and Vision, vol. 2, no. 1, pp. 1-104, 2007.[4] https://www.samsung.com/global/galaxy/galaxy-note8/.

[5] https://www.mindmedia.com/en/products/nexus-10-mkii/.

[6] https://www.accessdata.fda.gov/cdrh_docs/pdf5/K052489.pdf).

[7] https://www.ncbi.nlm.nih.gov/pubmed/9127780.

Reviewer 2 Report

The manuscript compares the assessment of heart rate from photoplethysmography (PPG) signals obtained with a smartphone (Samsung Galaxy Note8) in one finger and simultaneous PPG signals obtained with a PPG sensor in a NeXus device (in another finger from the opposite hand). The compared the heart rate assessed with both signals, which were processed offline to detect each pulse and measure the beat-to-beat interval from the maximum amplitude value (systolic peak). The manuscript is well written and the subject matter is of interest for readers of Sensors. However, there are many issues which should be addressed, as described below. Major issues 1) The title should be more precise. For instance “Monitoring of heart rate from photoplethysmographic signals using a Samsum Galaxy Note8 in underwater environments”. 2) Page 1, lines 24 to 26. Considering that it is known that underwater measurements are more difficult to undertake than measurements on air (dry conditions), dry conditions are the comparison medium and results should be mentioned in that way, for instance "...changing the medium from air to water decreases the PPG quality…" instead of "changing the medium from water to air improved the PPG quality. 3) Page 5, lines 238 to 255. The explanation of the effect of cold temperature in the PPG signal is more appropriate for the Introduction section. 4) Please review the statistical analysis. How was the normal distribution tested? When comparing more than two sub-groups (for instance dry and underwater environment with two different devices, as in Table 3), it is more appropriate to perform ANOVA for repeated measures with two factors (environment and device), with post-hoc comparisons adjusted by Bonferroni or another method. 5) Page 10, Figure 7. Review the figure legends, it appears that the panel (a) refers to the dry PPG signal and panel (b) is the simultaneous underwater PPG signal. Why the signal extracted from the Nexus device, panel (c), has a different time scale? 6) Page 10, Table 1 results and their description are very confusing. Are this RMS values (or amplitude values)? Why there is only one value for each temperature, device and condition? Was this assessed in only one person? What is the purpose of reporting the mean of amplitude from different temperatures? 7) Figure 8. Why is this represented with a boxplot (medians and percentiles) instead of another representation of mean and standard deviation (such as dot-error bar figure)? 8) Page 12 and 13 (Figure 9). The Bland-Altman plots are completely wrong. They should represent the mean of two simultaneous measurements (in the x-axis) versus the difference between the simultaneous measurements (in the y-axis). 9) Page 13 (last paragraph) and Figure 10. It is unclear how the "accuracy of measurement" was calculated. Minor issues 1) Page 2. Line 84. Define the FDA abbreviation. 2) Page 2, line 92. What is "dd"? 3) Page 2, line 93. Add “at each pulse (or heart beat)” so it reads: “… the maximum amplitude value at each pulse (or heart beat) to calculate the heart rate”. 4) Page 3, line 96 “evaluated” instead of “evaluate”. 5) The picture of the underwater environment (Figure 2b) should include the entire hand, to show a clearer description of the water immersion. 6) Page 5, lines 172 and 173. Please describe how the screenshots were taken (Was it done by pressing a button or tapping the screen? Who did the required action to take each screenshot?). 7) Page 11, Table 2. Move the label of dB units to the first column: “SNR (dB)” and delete it from the cells with the results. 8) Page 15, line 368. Change “In spit, of the fact…” to “In spite of the fact…”

Author Response

Reviewer: Reviewer 2

We would like to thank the reviewer for the positive judgment of our work and for the remarks, which have led to the improvement of our paper. Below we include a point-by-point reply to the comments. We have modified our paper following these comments and we hope that it is now suitable for publication. Please note that all changes to the text are highlighted in yellow.

General Comments & Questions:

The manuscript compares the assessment of heart rate from photoplethysmography (PPG) signals obtained with a smartphone (Samsung Galaxy Note8) in one finger and simultaneous PPG signals obtained with a PPG sensor in a NeXus device (in another finger from the opposite hand). The compared the heart rate assessed with both signals, which were processed offline to detect each pulse and measure the beat-to-beat interval from the maximum amplitude value (systolic peak). The manuscript is well written, and the subject matter is of interest for readers of Sensors. However, there are many issues which should be addressed, as described below.

RESPONSE: Thank you very much for your astute comments.The authors have clarified the points as the reviewer suggested and revised the manuscript.

  Specific Comments & Questions:

 (1)    Thetitle should be more precise. For instance, “Monitoring of heart rate from photoplethysmographic signals using a Samsung Galaxy Note8 in underwater environments”

RESPONSE: Thank you for your astute comments. As you suggested, the authors have changed the title into “Monitoring of heart rate from photoplethysmographic signals using a Samsung Galaxy Note8 in underwater environments” in the revised draft. 

(2)    Page1, lines 24 to 26. Considering that it is known that underwater measurements are more difficult to undertake than measurements on air (dry conditions), dry conditions are the comparison medium and results should be mentioned in that way, for instance "...changing the medium from air to water decreases the PPG quality…" instead of "changing the medium from water to air improved the PPG quality.

RESPONSE: Thank you very much for the astute comment. As you suggested, the authors have changed the sentence from “changing the medium from water to air improved the PPG quality” to “...changing the medium from air to water decreases the PPG quality…” as follows:

-----

[Lines 27-28, Page 1, Section Abstract] … changing the medium from air to water decreases the PPG quality, e.g., PPG signal amplitudes decrease from 0.560 to 0.112.

-----

(3)   Page5, lines 238 to 255. The explanation of the effect of cold temperature in the PPG signal is more appropriate for the Introduction section.

RESPONSE: Thank you for your comment.  As you suggested, the authors have added explanation about the effect of cold temperature in the PPG signal as follows: 

 -----

[Lines 90-101, Page 3, Section Introduction]: The effects of cold temperature on PPG signals have been studied in the viewpoints of 1) physiology and 2) sensor hardware. In the viewpoint of physiology, the effect of cold temperature on the PPG signals has been viewed with respect to blood vessel in our body [1-3]. In these study, temperature drop from 20 to 3  is observed to increase heart and respiration rate from 82 to 98 beat per minute and from 16 breath per minute [2, 3]. As a result, oxygen saturation level is also increased from 97% to 99%. Moreover the temperature drop causes blood viscosity to be increased, which decreases the amplitude PPG signal [4].On the other hand the effects of cold temperature on PPG signal quality has been studied in the viewpoint of sensor hardware including its photodiodes [4]. For example, temperature change from 25 to   shifts the wavelength by 18 nm, decreases the voltage by 0.1 volts and the current by 0.05 amps. However, cold temperature is observed not to change the pattern (or shape) of PPG signals in both viewpoints [5].

-----

(4)   Pleasereview the statistical analysis. How was the normal distribution tested? When comparing more than two sub-groups (for instance dry and underwater environment with two different devices, as in Table 3), it is more appropriate to perform ANOVA for repeated measures with two factors (environment and device), with post-hoc comparisons adjusted by Bonferroni or another method.

RESPONSE: Thank you very much for your astute comment. We checked that data were normally distributed using the Shapiro-Wilk test of normality. For the comparisons, we conducted two-way repeated measures ANOVA and we revised the manuscript based on the results as follow:

-----
[Lines 319-332, Page 12, Section Result]: Comparison of measurements of the two environments from the smartphone with the NeXus as a reference device was performed using the two-way repeated measures ANOVA; Environment (Dry, Underwater) and Device (Smartphone, NeXus). The results revealed that there was a significant interaction effect between Device and Environment (F (1,22) = 69.6, p<0.05). The interaction effect indicates that the heart rates from smartphone (Mean=80.52) were significantly higher than those from NeXus (Mean=76.39) in the underwater environment (Mean Difference=4.13, SE=1.84, p<0.05), while the heart rates from the smartphone (Mean=76.35) and NeXus (Mean=75.70) were not statistically different in the dry environment (Mean Difference=0.652, SE=0.59, p=0.29). 

Figure 8 shows the plot of each measurement’s amplitude for all volunteers between the dry and underwater environments. A paired-samples t-test was conducted to compare the measurement’s amplitude of heart rates in the dry and underwater environments. There was a significant difference in amplitude for the dry environment (Mean=0.37, STD=0.04) and the underwater environment (Mean=0.18, STD=0.04); t(22)=17.9, p<0.001.< span="">

 ----

[Figure 8, Page 12, Section Result]:

Figure 8.Amplitude plot: Left plot shows the amplitude of the dry environment from NeXus-10 MKⅡ and right plot shows the amplitude of underwater environment from Samsung Galaxy Note 8.

 -----

 (5)    Page10, Figure 7. Review the figure legends, it appears that the panel (a) refers to the dry PPG signal and panel (b) is the simultaneous underwater PPG signal. Why the signal extracted from the Nexus device, panel (c), has a different time scale?

RESPONSE: We apologize for this confusion. The authors have modified the figures (a), (b), and (c) for them to have the same time scale, and replaced the previous ones by these modified ones in the revised manuscripts as follows:

-----
[Figure 7, Pages 9-10, Section Results]:

           (a)

         (b)

(c)

Figure7.Representative example PPG signals measured in dry and underwater environments. a) PPG signal measured in underwater environment using Samsung Galaxy Note 8, b) PPG signal measured in dry environment using Samsung Galaxy Note 8, and c) PPG signal measured in dry environment using NeXus-10 MK II.

-----

 (6)    Page10, Table 1 results and their description are very confusing. Are this RMS values (or amplitude values)? Why there is only one value for each temperature, device and condition? Was this assessed in only one person? What is the purpose of reporting the mean of amplitude from different temperatures?

RESPONSE: We apologize for this confusion. All the values shown in Table 1 are not RMS values but average values. The averaged values of pulse amplitudes over 20 persons for each temperature, device, and condition are listed in Table 1 of the original draft. To make clear about each value, the authors have deleted Table 1 of the original draft, and added Table 2 in the revised draft which has all the values from 20 persons (Please see below). The reason why the authors report the mean of amplitude at different temperatures was to describe the effect of environment difference (dry vs. underwater environments) on the measured PPG signals. We have shown that the average amplitude decreases from 0.39 to 0.21 as the environment is changed from dry to wet at 18

----

 [Table 2, Page 11, Section Results]:

Table 2.Heart rate estimation, amplitude and SNR values from the NeXus-10 MK II device and Samsung Galaxy note 8 for dry and underwater environments at baseline temperature (18.

Dry Environment

Underwater Environment

Volunteer

 Gender

& Age

NeXus

Heart Rate

(bpm)

Smartphone

Heart Rate

(bpm)

Amplitude Value

(au)

SNR

(dB)

NeXus

Heart Rate

(bpm)

Samsung Heart Rate

(bpm)

Amplitude

Value

(au)

SNR

(dB)

1       

F 30

83

84

0.358

12.5

94

100

0.238

4.3

2        

M 32

76

76

0.356

12.9

77

84

0.204

4.2

3        

M 23

72

72

0.352

12.1

70

74

0.189

4.1

4        

F 37

95

94

0.322

11.8

88

92

0.184

5.2

5        

M 26

83

83

0.428

12.8

92

85

0.236

5.5

6        

M 29

71

72

0.383

12.6

68

73

0.241

5.3

7        

F 45

64

64

0.386

11.7

70

66

0.226

5.6

8        

M 27

77

78

0.327

12.1

70

73

0.177

4.2

9        

M 33

76

76

0.405

13.2

75

71

0.238

4.4

10     

M 26

63

63

0.323

13.1

64

68

0.178

4.2

11     

M 38

80

81

0.392

12.8

77

77

0.192

4.4

12     

M 25

63

63

0.391

13.8

69

65

0.187

4.6

13     

M 27

73

74

0.411

11.4

74

68

0.164

4.3

14     

F 72

92

92

0.315

14.1

96

91

0.168

4.8

15     

F 19

76

76

0.364

14.3

78

84

0.192

5.5

16     

F 21

65

65

0.292

14.5

63

69

0.115

5.1

17     

F 23

75

76

0.341

13.2

76

72

0.205

4.2

18     

F 22

74

74

0.313

12.9

72

80

0.215

4.4

19     

M 20

68

69

0.373

13.2

70

77

0.178

4.2

20     

M 19

86

86

0.364

13.1

85

94

0.113

4.5

Mean±STD

75.9±9

75.95±8.9

0.39±0.14

12.8 ± 1.4

75.4±9

80.2±10.2

0.2±0.1

4.9± 0.7

 (7)   Figure 8. Why is this represented with a boxplot (medians and percentiles) instead of another representation of mean and standard deviation (such as dot-error bar figure)?

RESPONSE: We apologize for this confusion. We checked that data were normally distributed using the Shapiro-Wilk test of normality. For the comparisons, we conducted two-way repeated measures ANOVA, and the significant interaction effect was further investigated using the tests of simple main effects as a post-hoc test. We revised the manuscript based on the results.

The authors have changed the plot representing the means and standard deviations in the revised manuscript as follows:

-----
[Lines 315-319, Page 12, Section Results]:We checked that data were normally distributed using the Shapiro-Wilk test of normality. For the comparisons, we conducted two-way repeated measures analysis of variance (ANOVA) using SPSS (IBM Corp. Released 2017. IBM SPSS Statistics for Windows, Version 25.0. Armonk, NY: IBM Corp), and the significant interaction effect was further investigated using the tests of simple main effects as a post-hoc test.

 Figure 8.Amplitude plot: Left plot shows the amplitude of the dry environment from NeXus-10 MKⅡ and right plot shows the amplitude of underwater environment from Samsung Galaxy Note 8.

-----

 (8)   Page 12 and 13 (Figure 9). The Bland-Altman plots are completely wrong. They should represent the mean of two simultaneous measurements (in the x-axis) versus the difference between the simultaneous measurements (in the y-axis)

RESPONSE: We apologize for this confusion. Thank you very much for your astute comment. The Bland – Altman plots have been corrected. The new Bland-Altman plots are shown below and replaced with the Bland-Altman plots in the paper. 

-----

[Lines 337-344, Pages 12-13, Section Results]: The Bland-Altman plots in figure 9 show the agreement between the heart rates obtained from the smartphone and NeXus in dry and underwater conditions. Figure 9a shows a bias between the mean differences of 0.65 in heart rates of smartphone and NeXus in dry condition with 95% limits of agreement interval of the mean differences. On the other hand, figure 9b shows a bias between the mean difference of 4.13 from smartphone in underwater and NeXus in dry condition, as well as an agreement interval with 95% of the mean differences. This implies that the bias between heart rates with smartphone and NeXus are larger in underwater condition than in dry condition. The difference of Samsung and NeXus comes from Samsung – Nexus.

[Figure 9, Page 13, Section Results]:

(a)

(b)

Figure 9. Bland-Altman plots of heart rate measurements after signal processing. a) Heart rate from Samsung Galaxy Note 8 and NeXus-10 MKⅡ in dry environment, b) Heart rate from NeXus-10 MKⅡ in dry and Samsung Galaxy Note 8 in wet environment.

-----

 (9)   Page 13 (last paragraph) and Figure 10. It is unclear how the "accuracy of measurement" was calculated.

RESPONSE: We apologize for this confusion. The accuracy is calculated using the following equations:

Accuracy(%) = 100% – 

 The authors have added these equations in the revised draft as follows:

 -----
[Lines 354-357, Page 13, Section Results]:

Accuracy is defined by the following equation:

Accuracy(%) = 100% – PE

where PE is the percentage of error and is derived by:

-----

Minor issues 

(10)  Page2. Line 84. Define the FDA abbreviation.

RESPONSE: Thank you for your astute comment.The authors have added the definition of FDA in the revised manuscript as follows:

-----
[Lines 89- 90, Page 2, Section Introduction]:Here, the heart rate obtained from NeXus-10 MKⅡdevice which has a Food and Drug Administration (FDA) approval, …

-----

 (11)  Page2, line 92. What is "dd"?

RESPONSE: We apologize for this confusionThe authors have fixed the mistype in the revised manuscript. The sentence “…and the dd to noise ratio (SNR).” has been changed to “… and the signal to noise ratio (SNR)” by the authors.

-----
[Line 175, Page 5, Section Method]: … and the signal to noise ratio (SNR).

 ----

(12)   Page 2, line 93. Add “at each pulse (or heart beat)” so it reads: “… the maximum amplitude value at each pulse (or heart beat) to calculate the heart rate”.

RESPONSE: Thank you for your astute comment. The authors have modified the sentence in the revised manuscript per your suggestion as follows. 

-----
[Line 176, Page 5, Section Method]: the maximum amplitude value at each pulse (or heart beat) to calculate the heart rate.

-----

(13)  Page 3, line 96 “evaluated” instead of “evaluate”.

RESPONSE: Thank you for your astute comment. The authors have changed the word “evaluate” to “evaluated” it in the revised manuscript.

 (14)  The picture of the underwater environment (Figure 2b) should include the entire hand, to show a clearer description of the water immersion.

RESPONSE: Thank you for your astute comment. As you suggested, the authors have replaced previous Figure 2 by new Figure 2 as below. The new Figure 2 in the revised draft includes the entire hand to show a clearer description of water immersion.

 -----

[Figure 2, Page 5, Section Materials]

(a)                                                                            (b)

Figure 2. Heart rate measurement in (a) dry and (b) underwater environments. Samsung Galaxy Note 8 PPG sensor is covered by the index finger of the right hand and the PPG sensor of NeXus-10 MKⅡ device is attached to the left hands index finger.

-----

 (15)  Page 5, lines 172 and 173. Please describe how the screenshots were taken (Was it done by pressing a button or tapping the screen? Who did the required action to take each screenshot?).

RESPONSE: We apologize for this confusion. The screenshots were taken by the Mobizen screen recording application. The screen recording application provides full HD recording at 60 fps. The screen shot recording was set up and the recording was started by an experimenter. We used a stitching technique to align consecutive screen shots after the screenshots were acquired. The authors have added this explanation in the revised manuscript as follows:

-----

[Lines 186-189, Page 5, Section Materials]: The screenshots were taken by the Mobizen screen recording application. The screen recording application provides full HD recording at 60 fps. We used a stitching technique to align consecutive screen shots after the screenshots were acquired.

 -----

 (16)  Page 11, Table 2. Move the label of dB units to the first column: “SNR (dB)” and delete it from the cells with the results.

RESPONSE: We apologize for this confusion. As you suggested, the authors have modified them in the revised as follows:

-----
[Table 1, Page 11, Section Results]: 

Table 1:Mean and standard deviation for the measurement duration and SNR in each environment.

Dry environment

 (Mean ± STD)

Underwater environment 

(Mean ± STD)

Time(s)

10 ± 2.7

20 ± 3.4

SNR (dB)

12.844 ± 1.42

4.962 ± 0.73

-----

 (17)   Page 15, line 368. Change “In spit, of the fact…” to “In spite of the fact…

RESPONSE: Thank you for your astute comment. The authors have corrected it in the revised draft as follows:

-----
[Line 390, Page 15, Section Conclusion]: In spite, of the fact that underwater …….

-----

Added References

[1]       T. E. Wilson and C. G. Crandall, "Effect of thermal stress on cardiac function," Exercise and sport sciences reviews, vol. 39, no. 1, p. 12, 2011.

[2]       G. E. White, S. G. Rhind, and G. D. Wells, "The effect of various cold-water immersion protocols on exercise-induced inflammatory response and functional recovery from high-intensity sprint exercise," European journal of applied physiology, vol. 114, no. 11, pp. 2353-2367, 2014.

[3]       J. Barcroft and F. Verzár, "The effect of exposure to cold on the pulse rate and respiration of man," The Journal of physiology, vol. 71, no. 4, pp. 373-380, 1931.

[4]       M. Khan, C. G. Pretty, A. C. Amies, R. Elliott, G. M. Shaw, and J. G. Chase, "Investigating the effects of temperature on photoplethysmography," IFAC-PapersOnLine,vol. 48, no. 20, pp. 360-365, 2015.

[5]       http://www.farnell.com/datasheets/2552940.pdf

Reviewer 3 Report

The paper is written in a clear language and it so easy to understand the main points.

I was though struggling to understand what is the key distinction of the work from a routine product testing against different environmental conditions. That is, I suppose, something that is usually included in the specifications.

Additionally, the introduction would benefit from explaining why underwater HR measurements is really needed. Are there cases, where a sensor would be used for constant monitoring even during underwater activities?

Author Response

We would like to thank the reviewer for the positive judgment of our work and for the remarks, which have led to the improvement of our paper. Below we include a point-by- point reply to the comments. We have modified our paper following these comments and we hope that it is now suitable for publication. The authors have clarified the points as the reviewer suggested and revised the manuscript. Please note that all changes to the original manuscript are highlighted in yellow.

Round  2

Reviewer 2 Report

The revised manuscript has improved. No more comments.

Reviewer 3 Report

Authors have considered the points raised in the first review.